# TreeX: Generating Global Graphical GNN Explanations via Critical Subtree Extraction

## Abstract

The growing demand for transparency and interpretability in critical domains has driven increased interests in comprehending the explainability of Message-Passing (MP) Graph Neural Networks (GNNs). Although substantial research efforts have been made to generate explanations for individual graph instances, identifying global explaining concepts for a GNN still poses great challenges, especially when concepts are desired in a graphical form on the dataset level. While most prior works treat GNNs as black boxes, in this paper, we propose to unbox GNNs by analyzing and extracting critical subtrees incurred by the inner workings of message passing, which correspond to critical subgraphs in the datasets. By aggregating subtrees in an embedding space with an efficient algorithm, which does not require complex subgraph matching or search, we can make intuitive graphical explanations for Message-Passing GNNs on local, class and global levels. We empirically show that our proposed approach not only generates clean subgraph concepts on a dataset level in contrast to existing global explaining methods which generate non-graphical rules (e.g., language or embeddings) as explanations, but it is also capable of providing explanations for individual instances with a comparable or even superior performance as compared to leading local-level GNN explainers.

## 1 Introduction

With the rapid advancements in artificial intelligence, the demand for transparent and explainable AI (XAI) has grown significantly. In sensitive domains like finance, security, and healthcare, where data is often structured as graphs, Message-Passing Graph Neural Networks (MPGNNs) have emerged as a prominent solution due to their straightforward design, remarkable effectiveness, and efficient performance. Understanding and explaining how they make predictions are essential to ensuring fairness, reliability, and maintaining control over AI tools on graph-structured data.

Existing works on GNN explainability can be categorized mainly into local-level explainability (Ying et al., 2019; Luo et al., 2020; Vu & Thai, 2020; Yuan et al., 2021; Shan et al., 2021; Bajaj et al., 2021; Lin et al., 2021; Wang et al., 2021; Feng et al., 2022; Xie et al., 2022; Zhang et al., 2022a; Lu et al., 2024b) and global-level explainability (Magister et al., 2021; Azzolin et al., 2023; Xuanyuan et al., 2023; Yuan et al., 2020; Wang & Shen, 2022; Huang et al., 2023). While local-level explainability focuses on identifying critical nodes, edges or subgraphs behind the GNN prediction on each individual graph instance, global-level explainability aims to provide a comprehensive explanation of a GNN's behavior on the entire dataset. However, current global-level GNN explanation methods either do not provide graphical explanations (Yuan et al., 2020; Magister et al., 2021; Wang & Shen, 2022) or cannot apply the extracted global rules to explaining individual data instances (Azzolin et al., 2023; Xuanyuan et al., 2023), making it hard to assess their faithfulness.

In this paper, we aim to solve the challenging problem of providing clear global explanations of GNNs on the dataset or class level in the format of behaviour differentiating subgraphs (rather than nodes, language rules or prototype embeddings), which is essential to molecular biochemistry and complex networks, e.g., to molecular property prediction and drug discovery. This is an unsolved challenge in the literature because there is an overwhelming complexity of enumerating and searching for subgraphs that are critical to GNN decisions.

Specifically, we propose sub**Tree**-based e**X**plainer (TreeX) (pronounced Trix), to unbox the inner mechanisms of MPGNNs by analyzing the *subtrees* generated through the *message-passing* process.

Figure 1: An illustrative example of the global explanations produced by TreeX and how the global explanations can be employed to explain individual instances. The global rule offers the optimal weights of different concepts to enhance the probability of predicting target class.

This approach yields intuitive subgraph concepts as global explanations, which can be straightforwardly employed to explain individual instances. We present an illustrative example of the global and local explanations produced by TreeX in Figure 1. We summarize our contributions as follows:

- **Subtree Extraction instead of Subgraph Search**: We perform subtree extraction to obtain subgraph concepts instead of traditional subgraph enumeration or search, reducing the search space per input graph from $n!$ to $n$, where $n$ is the number of nodes in a graph. This is due to the observation that there is a correspondence between $L$-layer subtrees incurred by GNN message passing and subgraphs.

- **Root Node Emebdding as Subtree Embedding**: For a $L$-layer GNN, each subtree incurred by message passing can be represented by its root node embedding at layer $L$. By representing subtrees using their root node embeddings, we not only avoid introducing another auxiliary mechanism to produce subgraph embeddings, but also convert another highly complex problem of identifying common structures among subgraphs into numerically clustering the subtrees in the embedding space, which can be done by off-the-shelf clustering methods. The rationale is that if the subtree embeddings are similar, they contribute to the GNN pooling layer similarly and their subgraph structures are also similar.

- **GNN explanations provided at both Global-Level and Local-Level**: Our proposed TreeX produces more intuitive subgraph concepts as global explanations, which are closer to the ground-truth crucial subgraphs and are more succinct and less noisy than state-of-the-art global-level baselines, suggesting better interpretability. Unlike existing global-level methods that do not explain individual graph samples, we also introduce how to utilize our extracted global explanations to identify critical interpretable structures in individual instances.

## 2 RELATED WORK

**Local-level GNN Explainability.** Post-hoc local-level, or instance-level, GNN explainability refers to the research problem of identifying crucial input nodes, edges, or subgraphs that significantly influence a GNN's prediction for a specific data instance. Most existing works in this domain view the target GNN as a black box. They typically design algorithms (Bui et al., 2024; Lu et al., 2024a; Feng et al., 2022; Yuan et al., 2021; Zhang et al., 2022b) or auxiliary models (Ying et al., 2019; Luo et al., 2020; Shan et al., 2021; Lin et al., 2021) to select a ratio of input nodes or edges, aiming to minimize loss on mutual information or fidelity performance. Some other works extend the concept of explaining neural networks from the vision domain to graph-structured data (Pope et al., 2019; Sundararajan et al., 2017; Yuan et al., 2022). However, they used to neglect the unique message-passing process of MPGNNs, which is essential for gaining deeper insights into the inner workings of these graph neural networks. More importantly, their explainability is limited to instances.

**Global-level GNN Explainability.** Post-hoc global-level, or model-level, GNN explainability is a relatively nascent research direction, with limited exploration and investigation. One line of existing global-level methods produce graph examples via either graph generation (Yuan et al., 2020; Wang & Shen, 2022; Nian et al., 2024) or graph edits (Huang et al., 2023) as the model-level explanations. While they generate numerous examples for each target class, they do not offer clear concepts, thus requiring human observers to draw conclusions (Xuanyuan et al., 2023; Kakkad et al., 2023). Another line of works provide factual global explanations by identifying important global concepts based on actual data (Kakkad et al., 2023). While subgraph explanations are gaining attention in local-level

explainability for their intuitive appeal (Yuan et al., 2021; Shan et al., 2021; Li et al., 2023; Zhang et al., 2022b), producing subgraphs as global concepts remains challenging due to the computational cost of searching among the possible subgraphs in the dataset. For example, GCExplainer (Magister et al., 2021) requires human-intervention to reduce the cost when determining the important subgraph concepts. These factual approaches provide explanations in various forms, such as boolean rules between latent clusters (Azzolin et al., 2023), and expert-defined natural language rules (Xuanyuan et al., 2023). However, these forms of explanations are less intuitive and understandable than graph concepts that our paper aims to extract. Another challenge with these methods is the quantitative assessment of the extracted global explanations, particularly when ground-truth explanations for the tasks are unavailable. This is because these approaches do not offer algorithms for applying global explanations to individual data instances, making it hard to measure their explanation fidelity.

## 3 PRELIMINARIES

**Graph Neural Networks.** Let $G = (V, E)$ be a graph with the associated nodes set $V$, edges set $E$, and $N = |V|$ represents the number of nodes. A GNN model $f(\mathbf{X}, \mathbf{A})$ maps the node features $\mathbf{X} \in \mathbb{R}^{N \times d}$ of dimension $d$ and the adjacency matrix $\mathbf{A} \in \mathbb{R}^{N \times N}$ indicating the existence or absence of edges $E$ to a target output, such as node labels, graph labels, or edge labels. Let $l$ be a message-passing layer in the GNN. At layer $l$, the GNN aggregates the neighbourhood information for each node $v \in V$ with the representation $\boldsymbol{h}_v^{(l-1)}$, and embeds the information into the next layer representation $\boldsymbol{h}_v^{(l)}$. In this paper, we focus on WL-based GNNs (or MPGNNs). Typical WL-based GNNs (Kipf & Welling, 2017; Xu et al., 2019; Hamilton et al., 2017) aggregate the information from the 1-hop neighbours $\mathcal{N}$ of $v$ as:

$$\boldsymbol{h}_v^{(l)} = \text{UPDATE}^{(l)}\left(\boldsymbol{h}_v^{(l-1)}, \text{AGG}^{(l)}\left(\left\{\boldsymbol{h}_u^{(l-1)} : u \in \mathcal{N}(v)\right\}\right)\right), \quad (1)$$

where $\text{UPDATE}^{(l)}$ and $\text{AGG}^{(l)}$ represent the updating and aggregation functions. In particular, an example maximally powerful MPGNN, GIN (Xu et al., 2019), updates node representations as:

$$\boldsymbol{h}_v^{(l)} = \text{MLP}^{(l)}\left(\left(1 + \epsilon^{(l)}\right) \cdot \boldsymbol{h}_v^{(l-1)} + \sum_{u \in \mathcal{N}(v)} \boldsymbol{h}_u^{(l-1)}\right). \quad (2)$$

**Subtrees.** Given a graph $G = (V, E)$, a *full $l$-hop subtree* $T_v^{(l)} = (V_{T_v^{(l)}}, E_{T_v^{(l)}})$ rooted at $v \in V$, is the entire underlying tree structure within $l$-hop distance from $v$, where $V_{T_v^{(l)}}, E_{T_v^{(l)}}$ are multisets, *i.e.*, a set with possibly repeating elements. Every element in $V_{T_v^{(l)}}$ is also an element in $V$; every element in $E_{T_v^{(l)}}$ is also an element in $E$. Figure 2 provides some examples of the full 2-hop rooted subtrees. The repetitions of the same node in the original graph are treated as distinct nodes in the subtrees, such that the pattern is still a cycle-free tree. Each subtree of $G$ corresponds to a subgraph in the original graph $G$. In the 1-WL test (Leman & Weisfeiler, 1968), subtrees are used to distinguish the underlying subgraphs.

## 4 PROPOSED METHOD

In this section, we propose a sub**Tree**-based e**X**plainer (TreeX) to explain MPGNNs. Overall, we mine over subtrees incurred by message-passing in GNN layers instead of enumerating subgraphs to reduce search space. As illustrated in Figure 2, TreeX consists of three phases: i) subtree-based local concept mining; ii) global concept extraction; iii) global rule generation for each class. In the main text of our paper, we focus on explaining the *maximally powerful MPGNNs*, whose last layer node embeddings are as expressive as the same layer 1-WL test (Leman & Weisfeiler, 1968). We move the analysis of explaining the less powerful MPGNNs to Appendix A.

### 4.1 SUBTREE-BASED EXPLAINER

**Local Concept Mining Based on Subtrees.** In the first phase, we aim to extract local subgraph concepts by mining over the rooted subtrees. Specifically, for a well-trained $L$ layer GNN, we consider

Figure 2: Overview of our proposed approach. This figure illustrates our approach for a 2 Layer GNN. The "subtrees" in this figure refer to the full $l$-hop subtrees. **Phase 1**: Collect subtrees in the graph, and extract local concept by identifying the overlapping substructures. **Phase 2**: Extract global concepts by clustering the local concepts. **Phase 3**: Generate global rules for each target class.

the last layer node embeddings as the representation of the corresponding full $L$-hop subtrees. We provide an analysis on the reason of doing so in Section 4.2. Then, we cluster these last layer node embeddings within the same graph. A typical choice of the clustering algorithm is the $k$-means algorithm (MacQueen et al., 1967; Forgy, 1965; MacKay, 2003), where given $k$ initialized clusters with centroids of $\boldsymbol{m}_1^{(0)}, \ldots, \boldsymbol{m}_k^{(0)}$ and a database, we assign each object $\boldsymbol{x}_p$ in the database to the cluster $S_i$, whose centroid is closest to it, based on the least squared Euclidean distance.

Next, for each local cluster $S_i$ where $i \in \{1, \ldots, k\}$ in each data instance $G = (V, E)$, we figure out the edges $\hat{E}_i$ covered by the most subtrees in $S_i$, and use them to induce the corresponding subgraph-level concept $G[\hat{E}_i]$ for this cluster. Formally, the count of an edge $e$ over the subtrees represented by the last layer node embeddings $\boldsymbol{x}_1, \ldots, \boldsymbol{x}_p, \ldots, \boldsymbol{x}_n$ in $S_i$, is calculated by:

$$\gamma(e|S_i) = |\{\boldsymbol{x}_p : e \in \psi(\boldsymbol{x}_p), \forall \boldsymbol{x}_p \in S_i\}|, \tag{3}$$

where $n$ is the number of subtrees in $S_i$, $\psi(\boldsymbol{x}_p)$ is the set of all the edges in the subtree represented by $\boldsymbol{x}_p$. Then $\hat{E}_i$ is defined as:

$$\hat{E}_i = \{e : \gamma(e|S_i) = max(\{\gamma(e_j|S_i), \forall e_j \in S_i\}), \forall e \in E\}. \tag{4}$$

Finally, we save the local graph concept $o_i = G[\hat{E}_i]$, and centroid $\boldsymbol{m}_{o_i}$ of $S_i$ as its embedding.

**Global Concept Extraction.** After the local concept mining phase across the entire dataset $\mathcal{D}$ with $|\mathcal{D}|$ data instances, we obtain a total of $k \cdot |\mathcal{D}|$ local graph concepts, where $k$ is the number of local clusters. In this subsection, we further cluster them to global-level concepts. In this phase, we utilize the $k$-means algorithm described previously due to its simplicity and efficacy in grouping objects.

We cluster the $k \cdot |\mathcal{D}|$ local concepts into $m$ global concepts $U_1, \ldots, U_m$. We save the local concept that is closest to the center of $U_j$ as the representative of it, and take the centroid $\boldsymbol{m}_{g_j}$ of $U_j$ as its embedding. Formally, the representative global concept $\hat{g}_j$ of the cluster $U_j$ is

$$\hat{g}_j = \text{argmin}_o \left\| \boldsymbol{m}_o - \boldsymbol{m}_{g_j} \right\|^2, \tag{5}$$

where $o$ is the local graph concept.

In the previous phase, local concept embeddings are determined by the centroids of local clusters, while representative local graph concepts are identified based on overlapping substructures. Consequently, multiple local concepts might share the same representative while having slightly different embeddings. This could result in them being grouped into different global clusters in the subsequent global clustering phase. To merge duplicated global concepts, we use an isomorphism test on the representative graph substructures of the global concepts. If two global concepts share the same graph representative, we merge them by averaging their embeddings. Following this merging process, we

ultimately obtain $\hat{m}$ global concepts in this phase. Given the WL test's effectiveness in determining graph isomorphism in most real-world cases (Zopf, 2022), we employ it for the isomorphism test in our experiments due to its efficiency.

**Global Rule Generation for Each Class.**    Now we have obtained $\hat{m}$ global graph concepts that are closely related to the GNN's prediction. However, the mapping between these global graph concepts and each output class remains unclear. In other words, we aim to determine how each global concept, either positively or negatively, influences each specific output class.

To achieve this goal, we feed the trainable weighted sum of the embeddings of the $\hat{m}$ global graph concepts to the original classifier of GNNs, and optimize for the trainable weights at each output class, respectively. Formally, consider a well-trained GNN $f(\cdot) = \Phi \circ \text{READOUT} \circ \Psi(\cdot)$, where $\Phi$ refers to a $L$-layer message-passing module, $\Psi$ refers to a classifier module, the classifier prediction on the weighted global concepts at instance $i$ is

$$\hat{\boldsymbol{y}}_i = \Psi\left(\boldsymbol{w}_t \mathbf{K}_i \mathbf{M}\right), \tag{6}$$

in which $\boldsymbol{w}_t$ is the trainable vector of length $\hat{m}$, $\mathbf{K}_i \in \mathbb{N}^m$ refers to the number of subtrees that contain each of the global concepts in the data instance $G_i$, $\mathbf{M}$ refers to the global concepts for $\mathbf{M} = \text{STACK}\left(\boldsymbol{m}_{g_1}, \ldots, \boldsymbol{m}_{g_{\hat{m}}}\right)$. We then aim to optimize the negative log likelihood (NLL) loss with a $L2$-penalty on $\boldsymbol{w}_t$:

$$\mathcal{L}(\hat{\boldsymbol{y}}_i, y_t) = -\log \frac{\exp(\hat{\boldsymbol{y}}_{y_t})}{\sum_{c=1}^{C} \exp(\hat{\boldsymbol{y}}_c)} + \lambda \left\|\boldsymbol{w}_t\right\|_2, \tag{7}$$

where $y_t$ refers to the target class, $\lambda$ is a weighing factor. We incorporate a penalty term for the following reason. As previously mentioned, $\boldsymbol{w}_t \mathbf{K}_i \mathbf{M}$ represents the weighted sum of the global graph concepts. By controlling the overall weights, our objective is to encourage the critical concepts to occupy only a minor portion of the "dataset embedding", which can mimic the READOUT functions typically employed in GNNs. In order words, we introduce this penalty term based on the intuition that GNNs can effectively represent GNNs at each target class using a limited number of significant concepts, with the remaining non-significant substructures exerting minimal influence on the prediction of the target class.

## 4.2    ANALYSIS OF CRITICAL SUBTREE-BASED DESIGNS

The framework we introduced in Section 4.1 utilizes two critical designs: i) mining over *subtrees* rather than subgraphs to extract concepts; ii) representing the full $L$-hop subtrees by the $L$-th layer root node embeddings.

The first design, sharply reduces the search space, compared with mining over all possible subgraphs. This is because in each graph instance with $N$ nodes, there are exactly $N$ full $L$-hop subtrees, while up to $N!$ possible subgraphs. Although searching over possible subgraphs in a single instance is feasible in local-level explainability (Yuan et al., 2021; Zhang et al., 2022b; Shan et al., 2021), it becomes much more challenging in global-level explainability, where the dataset can be large. Our TreeX, mining over subtrees, provides a practical direction for extracting global graph concepts.

The second design further improves the mining process compared with existing explainability methods (Yuan et al., 2021; Azzolin et al., 2023). Typically, existing methods represent subgraphs by feeding them into the original GNN to obtain subgraph embeddings, necessitating additional calculations for each possible subgraph. In contrast, we directly use the node embeddings to represent subtrees, which are easily obtained by feeding the graph to the GNN just once. In the remainder of this section, we explain why we can use root node embeddings to represent subtrees, by showing that the $l$-th layer root node embedding from a maximally powerful MPGNN is an exact mapping of the corresponding full $l$-hop rooted subtree.

**Definition 4.1 (Perfect Rooted Tree Representation).** Let $T_v$ denote a tree in a countable space $\mathcal{X}$, which is rooted at node $v$, $f(\cdot)$ be the only function to generate the presentations of rooted trees in the space, $\boldsymbol{h}_v$ be the representation of $T_v$. We define $\boldsymbol{h}_v$ be the Perfect Rooted Tree Representation of $T_v$, if the following holds: For any arbitrary same-depth rooted tree $T_u$ in the same countable space, $\boldsymbol{h}_v = \boldsymbol{h}_u$ if and only if $T_v$ is isomorphic to $T_u$.

We then show that if both AGG and UPDATE in Equation (1) are injective, then the $l$-th layer node embedding is a Perfect Rooted Tree Representation of the full $l$-hop rooted subtree by mathematical

induction, which is described in Theorem 4.2. The proofs of all the lemmas and theorems can be found in Appendix B.

**Theorem 4.2.** *Given a graph $G = (V, E)$ with the countable input node features $\mathbf{X}$, and a L-layer GNN $f(\cdot)$ that updates the layer-wise node-embeddings by Equation (1). Then $\forall l \in \{1, \ldots L\}$ and $\forall v \in V$, the l-th layer node embedding $\mathbf{h}_v^{(l)}$ is a Perfect Rooted Tree Representation of the full l-hop subtree rooted at $v$, if the functions* AGG *and* UPDATE *in Equation* (1) *are injective.*

As presented in Equation (2), the maximally powerful MPGNN, e.g., GIN, utilizes add-pooling and MLP as the AGG and UPDATE functions, which are both injective for countable inputs. By Theorem 4.2, we acquire that GIN uniquely maps the full $l$-hop subtrees to the $l$-th layer embeddings of their root nodes. Therefore, we can directly use the root node embeddings to represent the corresponding rooted subtrees. Due to the space limit, we have moved the discussion of representing subtrees by the less powerful MPGNNs in Appendix A.1.

### 4.3 EXPLAINING LOCAL INSTANCES WITH GLOBAL RULES

In earlier sections, we discussed how we generate global rules for MPGNNs at the target class via a weighted sum of global subgraph concepts. In this section, we introduce how to identify critical input subgraphs for each individual instance with the global rules generated by our method.

For a specific data sample, we first perform local subtree mining on it like we did in Section 4.1, to obtain the local concepts in the data instance. Following this, we calculate the distance from each local concept to the previously extracted global concepts and fit the local concepts in this instance to the nearest global concepts. Then, we can construct a concept mask $K \in \mathbb{R}^m$ for this data instance, where, similar to Section 4.1, $K$ indicates the numbers of subtrees in the concepts, $m$ refers to the total number of global concepts in the global rule. The importance $I_t$ of the concepts in this data instance for the target class $y_t$ can be calculated by $I_t = K\boldsymbol{w}_t$, where $\boldsymbol{w}_t \in \mathbb{R}^m$ is the trained weight for all concepts in the global rule for class $y_t$. The concepts that do not exist in the data instance will naturally receive zero importance as shown in Figure 1.

The explanations we produce are class-specific, meaning that for each instance, we offer a weighted sum of the global concepts aimed at maximizing the prediction probability for *each specific class*. As a result, we not only can explain the instances that are correctly predicted by GNNs, but can also discover the cause of incorrect predictions. That is, we can investigate the importance of various concepts for both the wrong class and the true class for a wrongly predicted data instance. Analyzing the explanation for the wrong class can give us the information about which concepts make the original GNN gain more confidence in predicting the wrong class, hence becomes less confident in predicting the correct class. Analyzing the explanation for the true class can give us the information about which critical concepts are overlooked by the original GNN, thus leading it to predict the incorrect class.

## 5 EXPERIMENTS

In this section, we first compare the global explanations produced by TreeX with those from existing global-level approaches, illustrating that TreeX can generate clear subgraph concepts while existing methods cannot. Second, we assess the faithfulness of TreeX by comparing its fidelity with that of leading local-level methods. Note that fidelity is a metric calculated on individual instances, so we cannot compute fidelity for existing global-level methods, as they are unable to explain local instances like TreeX. Third, we demonstrate how TreeX provides insights into the reasons for incorrect GNN predictions. Finally, we offer an efficiency analysis of TreeX.

### 5.1 EXPERIMENTAL SETUP

**Datasets.** Similar to prior works (Azzolin et al., 2023; Xuanyuan et al., 2023), we mainly focus on graph classifications and conduct experiments on two synthetic datasets BA-2Motifs (Luo et al., 2020) and BAMultiShapes (Azzolin et al., 2023), as well as two real-world datasets Mutagenicity (Kazius et al., 2005) and NCI1 (Wale et al., 2008; Pires et al., 2015) to demonstrate the efficacy of our approach. The statistics of these datasets are in Appendix C. The BA-2Motifs and the BAMultiShapes

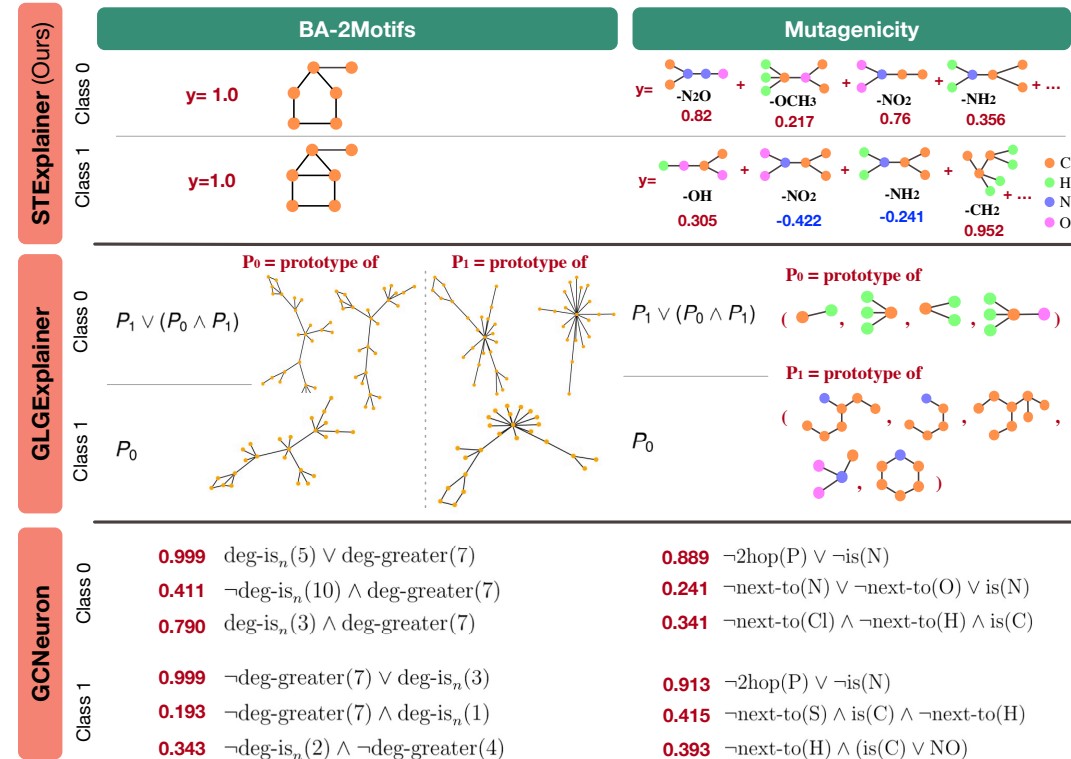

Figure 3: Global explanations by TreeX (ours), GCNeuron and GLGExplainer. We run both baseline methods so that they explain the same GNN models as our approach. Due to space limit, the explanations on BAMultiShapes and NCI1 datasets are moved to the appendix.

datasets employ Barabasi-Albert (BA) graphs as base graphs. In the he BA-2Motifs dataset, Class 0 graphs are augmented with five-node cycle motifs, while Class 1 graphs are enriched with "house" motifs. In the BAMultiShapes dataset, the graphs contain randomly positioned house, grid, and wheel motifs. Class 0 includes plain BA graphs and those with individual motifs or a combination of all three. In contrast, Class 1 comprises BA graphs enriched with any two of the three motifs. Mutagenicity and NCI1 are real-world chemical and medical datasets, which are challenging for both classification or explainability due to their complex graph structures. In the Mutagenicity dataset, graphs in Class 0 are mutagenic molecules, and graphs in Class 1 are non-mutagenic molecules. NCI1 contains a few thousand chemical compounds screened for activity against non-small cell lung cancer and ovarian cancer cell lines. The intricacy of these datasets makes it challenging to derive definitive classification rules, even for human experts.

**Evaluation Metric.** We use both prediction accuracy fidelity (Yeh et al., 2019; Zhou et al., 2021) and prediction probability fidelity (Yuan et al., 2022) to evaluate the faithfulness of our method in explaining model predictions on individual instances. They are respectively defined as $AccFidelity = \frac{1}{N}\sum_{i=1}^{N} \mathbb{1}(\hat{y}_i = y_i)$ and $ProbFidelity = \frac{1}{N}\sum_{i=1}^{N} f_{y_i}(G_i) - f_{y_i}(G_i^{\mathcal{X}})$, where $y_i$ refers to GNN's prediction on the original input, $\hat{y}_i$ refers to the prediction on the explanation, $G_i$ refers to the original input, $G_i^{\mathcal{X}}$ refers to the explanation.

**Baselines.** We qualitatively compare with two recent factual global-level explainers GLGExplainer (Azzolin et al., 2023) and GCNeuron (Xuanyuan et al., 2023). We also quantitatively compare the explanation fidelity of our approaches on the local instances with the leading local-level baselines such as GradCAM (Pope et al., 2019), GNNExplainer (Ying et al., 2019), PGExplainer (Luo et al., 2020), SubgraphX (Yuan et al., 2021), EiG-Search (Lu et al., 2024a).

**Implementation Details.** Please see Appendix C and D.2 for details.

Table 1: Prediction *accuracy* fidelity ($AccFidelity$ ($\uparrow$)) performance of TreeX and the local-level baselines on the BA-2Motifs, BAMultiShapes, Mutagenicity, NCI1 test datasets.

| Method | BA-2Motifs | | BAMultiShapes | | Mutagenicity | | NCI1 | |
|---|---|---|---|---|---|---|---|---|
| | Class 0 ($\uparrow$) | Class 1 ($\uparrow$) | Class 0 ($\uparrow$) | Class 1 ($\uparrow$) | Class 0 ($\uparrow$) | Class 1 ($\uparrow$) | Class 0 ($\uparrow$) | Class 1 ($\uparrow$) |
| GradCAM (Pope et al., 2019) | **1.00**±**0.00** | 0.92±0.00 | **1.00**±**0.00** | 0.00±0.00 | 0.81±0.00 | 0.77±0.02 | 0.94±0.00 | 0.16±0.00 |
| GNNExplainer (Ying et al., 2019) | 0.92±0.05 | 0.82±0.07 | 0.92±0.03 | 0.54±0.13 | 0.83±0.02 | 0.81±0.03 | 0.89±0.02 | 0.64±0.17 |
| PGExplainer (Luo et al., 2020) | 0.21±0.19 | 0.98±0.02 | 0.99±0.02 | 0.05±0.10 | 0.93±0.02 | 0.16±0.23 | 0.88±0.05 | 0.15±0.37 |
| SubgraphX (Yuan et al., 2021) | 0.56±0.12 | 0.90±0.08 | 0.99±0.01 | 0.04±0.07 | 1.00±0.01 | 0.45±0.16 | 0.99±0.01 | 0.07±0.28 |
| EiG-Search (Lu et al., 2024a) | **1.00**±**0.00** | **1.00**±**0.00** | **1.00**±**0.00** | 0.98±0.00 | 0.99±0.00 | **0.94**±**0.00** | **0.99**±**0.00** | 0.68±0.00 |
| TreeX (ours) | **1.00**±**0.00** | **1.00**±**0.00** | **1.00**±**0.00** | **0.99**±**0.01** | **1.00**±**0.00** | 0.93±0.05 | 0.94±0.05 | **1.00**±**0.00** |

Table 2: Prediction *probability* fidelity ($ProbFidelity$ ($\downarrow$)) performance of TreeX and the local-level baselines on the BA-2Motifs, BAMultiShapes, Mutagenicity, NCI1 test datasets.

| Method | BA-2Motifs | | BAMultiShapes | | Mutagenicity | | NCI1 | |
|---|---|---|---|---|---|---|---|---|
| | Class 0 ($\downarrow$) | Class 1 ($\downarrow$) | Class 0 ($\downarrow$) | Class 1 ($\downarrow$) | Class 0 ($\downarrow$) | Class 1 ($\downarrow$) | Class 0 ($\downarrow$) | Class 1 ($\downarrow$) |
| GradCAM (Pope et al., 2019) | **0.00**±**0.00** | 0.08±0.00 | 0.00±0.00 | 0.97±0.00 | 0.06±0.00 | 0.12±0.00 | 0.01±0.00 | 0.74±0.00 |
| GNNExplainer (Ying et al., 2019) | 0.02±0.00 | 0.17±0.05 | 0.04±0.01 | 0.17±0.05 | 0.08±0.03 | 0.13±0.06 | 0.07±0.02 | 0.27±0.05 |
| PGExplainer (Luo et al., 2020) | 0.82±0.14 | 0.03±0.05 | 0.00±0.01 | 0.97±0.02 | 0.20±0.03 | 0.40±0.21 | 0.08±0.01 | 0.81±0.05 |
| SubgraphX (Yuan et al., 2021) | 0.16±0.06 | 0.00±0.04 | **-0.01**±**0.00** | 0.87±0.02 | -0.03±0.02 | 0.31±0.04 | -0.02±0.01 | 0.40±0.03 |
| EiG-Search (Lu et al., 2024a) | **0.00**±**0.00** | **0.00**±**0.00** | **-0.01**±**0.00** | 0.01±0.00 | -0.12±0.00 | 0.01±0.00 | -0.05±0.00 | 0.27±0.00 |
| TreeX (ours) | **0.00**±**0.00** | **0.00**±**0.00** | -0.01±0.01 | **0.03**±**0.01** | **-0.25**±**0.03** | **-0.21**±**0.04** | **-0.08**±**0.02** | **-0.14**±**0.03** |

## 5.2 RESULTS

**Comparison with Other Global Explainers.** As summarized in Section 2, existing factual global-level explainers produce explanations in different forms. We illustrate the global explanations produced by our approach, and two existing factual global-level baselines in Figure 3. As shown in this figure, GLGExplainer has limitations in delivering clear global explanations. This is because they generate latent vectors as the prototypes, where they provide several random local explanations for examples within the latent cluster, lacking clear motifs to represent each prototype. Therefore, from the perspective of providing intuitive and clear explanations, their global-level explanations remain implicit and require human experts to interpret and draw meaningful conclusions.

On the other hand, GCNeuron provides global explanations in the form of logical rules with human-defined premises. However, without prior knowledge, it becomes challenging to define meaningful natural language rules as premises when dealing with the BA-2Motifs dataset. Consequently, their explanations rely on the abstract concepts like the "degree of nodes" or "degree of neighboring nodes", which can make them quite perplexing and challenging for humans to understand. When applied to the Mutagenicity dataset, GCNeuron manually defines 44 premises, including terms like "NO2", "NO", "is(C)", "neighbour of C", "2-hop from C". However, GCNeuron fails to recognize "NO2" as a Class 0 motif, even though it's known to be relevant to mutagenic effects (Luo et al., 2020).

Conversely, our TreeX is able to accurately extract the critical global motifs on BA-2Motifs mentioned in Section 5.1. The global explanation produced by TreeX for this dataset indicates that: if a graph contain the five-node cycle motif, then it is a Class 0 graph; if a graph contain the "house" motif , then it is a Class 1 graph. Other substructures are not important. On the Mutagenicity dataset, TreeX successfully identifies the "-NO2" and "-NH2" chemical groups as Class 0 patterns, which are well-known to be related to the mutagenic effect of molecules, as discussed in previous studies (Ying et al., 2019; Luo et al., 2020; Debnath et al., 1991). Additionally, it identifies "-N2O", "-OCH3" as the Class 0 motifs, and "-CH2", "-OH" as the Class 1 motifs, albeit with relatively lower but still positive confidence. These chemical groups have been widely studied in terms of their mutagenic effects (Hill et al., 1998; Baden & Kundomal, 1987). Explicitly highlighting these chemical groups provides a more comprehensive understanding of how GNNs make decisions and can be valuable for debugging GNNs.

Due to the space limit, we have moved the comparisons on the BAMultiShapes and NCI1 datasets to appendices. Please see Appendix D.1 for more results and discussions on these datasets.

It is worth-noting that as we discussed in Section 2, since these existing global-level approaches do not offer algorithms for employing their extracted global explanations to the data instances in the test set, we do not access the explanation fidelity of them.

Table 3: Evaluation of TreeX in uncovering the cause of on the incorrect predictions. Note that the prediction accuracy of these falsely predicted data instances is 0.

| Dataset | BAMultiShapes | Mutagenicity | NCI1 |
|---------|---------------|--------------|------|
| | Prediction Accuracy ($Acc\,(\uparrow)$) | | |
| Class 0 | 1.00±0.00 | 1.00±0.00 | 0.98±0.02 |
| Class 1 | 1.00±0.00 | 0.90±0.05 | 0.74±0.08 |

Figure 4: Visualization of employing the global explanations produced by TreeX to discover the cause of the incorrect prediction of the GNN. Due to the space limit, we omit the concepts that are not in this graph.

**Faithfulness of TreeX.** The results of the prediction accuracy fidelity and probability fidelity are presented in Table 1 and Table 2 respectively, where we report the standard deviation over 5 runs. In both tables, we utilize the explainers to only explain the data instances that are correctly predicted by the GNNs. It is observed that TreeX achieves nearly optimal prediction accuracy fidelity and outstanding prediction probability fidelity performance across all the datasets, while the baselines struggle to extract explanations meeting with GNN's predictions. This shows that the global explanations produced by our approach correctly match the behaviour of the GNNs. Additionally, TreeX achieves even better $ProbFidelity$ performance on the Mutagenicity and NCI1 datasets than the state-of-the-art local-level explainer EiG-Search. It is because unlike the baselines that simply identify the explanation subgraphs, our approach additionally learns the best weights of them for more optimized prediction probability. Notably, a lower $ProbFidelity$ smaller than 0 means that the GNN is more confident in predicting the target classes based on the explanations than on the original graphs. This emphasizes the exceptional ability of TreeX to highlight the critical concepts for the target classes and produce faithful explanations.

**Discover Reasons for Incorrect GNN Predictions.** As we discussed in Section 4.3, unlike many existing GNN explainability methods that solely focus on identifying which input substructures lead to GNN's predictions, our class-specific approach also explains why the GNN does not predict other classes. That is, we provide some insights on how a GNN might be improved. TreeX aims to globally provide users with an understanding of why the GNN makes correct predictions on some samples and incorrect predictions on other samples. The global explanation produced by TreeX for a target class is a set of subgraph concepts with their weights. Positive weights indicate that the GNN relies on these subgraph concepts to predict the target class, while negative weights imply that the existence of these subgraph concepts makes the GNN less confident in predicting the target class. We provide two example global explanations for both Class 0 and Class 1 in Figure 4.

Following the global explanations, we can discover the reasons behind the incorrect predictions. For example, for a data sample in Figure 4, the true label is 1, while the given GNN predicts 0. By examining the existence of the subgraph concepts in this sample, we find that the GNN predicts Class 0 and is less confident in predicting Class 1 because it has Concept 2. We could also provide an insight to the developer of the GNN that if the GNN can be improved either through neural parameters or the design of the pooling mechanism, by increasing its interest in Concept 1 for Class 1, it may be able to correct its prediction on this data sample.

In Table 3, we further report the rate of predicting the true labels using our extracted global explanations of the corresponding classes, for the incorrect predictions of the original GNNs. The results show that for most of the GNN's incorrect predictions, our approach is able to show how adjusting weights of various concepts can help to correct the predictions, highlighting the effectiveness of our approach in uncovering causes of incorrect predictions.

Table 4: Empirical efficiency comparison of TreeX (global) with and local-level methods. The elapsed time is reported as the average per data instance.

| Dataset | TreeX (global) | SubgraphX (local) | EiG-Search (local) |
|---|---|---|---|
| BA-2Motifs | **0.08s** | 63.7s | 0.09s |
| Mutagenicity | **0.14s** | 419.8s | 0.14s |

**Visualization of Concept Scores on Instances.** Please see Appendix D.3 for examples of visualizing our explanations on individual instances.

### 5.3 TIME ANALYSIS

The main focus of our work is to generate clear *subgraph concepts* as *global* explanations for GNNs at the dataset and task levels. Although some approaches can explain GNNs at the global or model level, they do not produce clear subgraph concepts, as discussed in Section 2 and demonstrated in Figure 3. Therefore, we refrain from efficiency comparisons with those existing global-level approaches.

On the other hand, several local-level explanation methods attempt to extract critical subgraphs for individual instances (Bui et al., 2024; Lu et al., 2024a; Zhang et al., 2022b; Yuan et al., 2021). While these methods produce clear subgraph concepts, their explainability is limited to the instance level, rather than the dataset or task level. Moreover, it is challenging to extract global subgraph concepts from the local critical subgraphs generated by these methods. This is because local-level critical subgraphs can be noisy, containing redundant substructures that are only crucial for certain local predictions and not part of the core concepts at the global or task level. Removing such redundancy to produce global-level subgraph concepts may require costly *subgraph matching*, which is known to be NP-hard. In contrast, our approach removes redundant substructures by pruning the local subtrees, as discussed in Section 4.1. The complexity of this redundancy removal process is less than $O(enk)$, much more efficient than the subgraph matching, where $n$ and $e$ are the number of nodes and edges in each graph, and $k$ is the number of local clusters.

Although local-level explanation approaches are not capable of producing global subgraph concepts like our TreeX, we still perform an efficiency comparison of concept extraction between our method and theirs. In Table 4, we compare the average elapsed time for explaining each data instance on the BA-2Motifs and Mutagenicity datasets with two prominent local subgraph-level baselines, SubgraphX (Yuan et al., 2021) and EiG-Search (Lu et al., 2024a). For the baselines, we use their methods to explain all instances in the datasets and calculate the average elapsed time. For our method, we first obtain the global explanations as discussed in Section 4.1, then apply the algorithm described in Section 4.3 to produce local explanations for all instances, and finally divide the total elapsed time by the number of instances.

In Table 4, we can observe that TreeX is much more efficient than SubgraphX, and as efficient as EiG-Search, even though the latter ones only provides local-level graph explanations per graph instance, while globally it is not able to generate behavior-defining subgraphs for the entire dataset or task. In fact, TreeX has a more robust and stable local-level fidelity performance across different benchmarks while the performance of GradCAM and EiG-Search may vary across benchmarks. This again implies the value of further investigating and obtaining clean, less noisy and consistent global graph explanations rather than per-instance explanations that existing works focus on.

## 6 CONCLUSION

In this paper, we introduce a novel approach TreeX for explaining MPGNNs from the perspective of their distinct message-passing process. We extract intuitive and clear subgraph concepts by mining over the full-$L$-hop subtrees at each graph instance, unlike existing approaches that identify latent prototypes or human-defined rules. We utilize the last-layer node embeddings to help represent the concepts, thereby avoiding additional complex calculations of the concept embeddings. Due to this design, we are able to employ the global explanations extracted by our method to explain the individual instances, while the existing global-level baselines fail to do so. Moreover, unlike many existing GNN explaining approaches that solely focus on explaining the correct predictions of GNNs, our approach offers insights of the causes of incorrect predictions. In the future, GNNs may be refined with these insights to improve their classification performance.

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

APPENDIX

# A EXPLAINING LESS POWERFUL MPGNNS

The 1-WL test is able to decide the graph isomorphism in most real-world cases (Zopf, 2022). Therefore, for the maximally expressive MPGNNs, we induce the subgraph-level concepts with the full $l$-hop subtrees in Section 4.1. However, there exist many GNNs like GCN (Kipf & Welling, 2017) and GraphSAGE (Hamilton et al., 2017) that are significantly less expressive than the 1-WL test. In this subsection, we explain how these less expressive GNNs can be explained using our proposed TreeX.

GCN (Kipf & Welling, 2017) updates node representations as:

$$\boldsymbol{h}_v^{(l)} = \text{RELU}\left(W \cdot \text{MEAN}\left\{\boldsymbol{h}_u^{(l-1)}, \forall u \in \mathcal{N}(v) \cup \{v\}\right\}\right), \tag{8}$$

where $W$ is a learnable matrix, and MEAN represents the element-wise mean-pooling.

GraphSAGE (Hamilton et al., 2017) updates node representations as the linear mapping on the concatenation of the last-layer node embedding and the aggregation of the neighbouring node embeddings:

$$\boldsymbol{h}_v^{(l)} = \sigma\left(W \cdot \left[\boldsymbol{h}_v^{(l-1)}, \text{MAX}\left(\left\{\sigma\left(W \cdot \boldsymbol{h}_u^{(l-1)}\right), \forall u \in \mathcal{N}(v)\right\}\right)\right]\right), \tag{9}$$

where MAX represents the element-wise max-pooling, $\sigma$ refers to RELU.

## A.1 RELATIONSHIP BETWEEN SUBTREES AND NODE EMBEDDINGS FOR LESS POWERFUL MPGNNS

In Section 4.2, we have studied the node representations of the most expressive MPGNNs, i.e., the ones as expressive as the 1-WL test, which is reviewed in Appendix B.1. In this subsection, we discuss the less expressive GNNs by study the UPDATE and AGG functions of them.

If the UPDATE function is injective, then the distinctness of the embeddings will not change after feeding into the UPDATE function. Therefore, if a $l$-th layer node embedding after AGG can maps two distinct subtrees at the same time, then after the injective UPDATE function, it will represent the same pair of distinct subtrees. This forms the following corollary.

**Corollary A.1.** *Given a graph $G = (V, E)$, let $f(\cdot)$ denote a $L$-layer GNN that updates the countable layer-wise node-embeddings by Equation (1). We define the intermediate $l$-th layer embedding derived by the AGG function of node $v \in V$ as $\boldsymbol{h}_{v,\text{AGG}}^{(l)}$, where $\boldsymbol{h}_v^{(l)} = \text{UPDATE}^{(l)}\left(\boldsymbol{h}_{v,\text{AGG}}^{(l)}\right)$. Then the following holds:*

*(i) If UPDATE is injective, then $\boldsymbol{h}_v^{(l)}$ is a Perfect Rooted Tree Representation of the full $l$-hop subtree rooted at $v$ if and only if $\boldsymbol{h}_{v,\text{AGG}}^{(l)}$ is a Perfect Rooted Tree Representation of the full $l$-hop subtree rooted at $v$.*

*(ii) If UPDATE is injective, then $\boldsymbol{h}_v^{(l)}$ is a mapping of both a full $l$-hop subtree and another non-isomorphic $l$-hop subtree rooted at $v$, if and only if $\boldsymbol{h}_v^{(l)}$ is a mapping of the same pair of non-isomorphic subtrees at the same time.*

*Proof.* Please see Appendix B.3 for the proof. $\square$

In the family of WL-based GNNs, a 1-layer MLP (with bias term) or a 2-layer MLP is typically used as the UPDATE function, which are both injective functions. Some variants of GNNs, like GraphSAGE (Hamilton et al., 2017), may additionally utilize a concatenation as shown in Equation (9), which is also injective. Therefore, we assume the UPDATE function in Equation (1) of less powerful MPGNNs is a injective function.

Next, we discuss the AGG function in Equation (1), including the commonly used add-pooling, mean-pooling and max-pooling methods. The most expressive add-pooling is discussed in Section 4.2.

Mean-pooling, as investigated in (Xu et al., 2019), captures the "distributions" of elements in a multiset. In other words, there may exist another subtree $T_v^{(l)\prime}$ that contain the same set of unique elements as the full $l$-hop subtree $T_v^{(l)}$, where the distribution of unique elements in $T_v^{(l)\prime}$ is the same as in $T_v^{(l)}$. Such two subtrees will have the same root node embedding if using the mean-pooling, which can be treated as a perfect representation of the *node distribution* in the full $l$-hop rooted subgraphs. GCN is an example of using mean-pooling as shown in Equation (8).

Max-pooling treats a multiset as a set (Xu et al., 2019). This means if there exists two subtrees $T_v^{(l)\prime}$ and $T_v^{(l)}$ contain same set of unique elements, they will have the same root node embedding, which can be treated as a perfect representation of the unique node set in the full $l$-hop rooted subgraphs. GraphSAGE is an example of max-pooling as shown in Equation (9).

It is worth noting that the MPGNNs with mean-pooling or max-pooling tend to be less expressive hence are less preferred for most tasks. Therefore, our primary focus in this paper is to explain GNNs that demonstrate expressiveness comparable to the WL-test algorithm. In these GNNs, node embeddings precisely represent the full $l$-hop subtrees.

## A.2 HASH MODEL TO EXPLAIN THE LESS POWERFUL MPGNNS

As we discussed in Appendix A.1, GNNs that update the node embeddings by mean-pooling may produce the same root node embeddings for the subtrees with the same node distribution, and the ones using max-pooling may produce the same root node embeddings for the subtrees with the same node set. For these GNNs, there is a higher risk of clustering multiple entirely different substructures to the same concept as those non-isomorphic subtrees may share the same root node embedding.

To mitigate this issue, we introduce a *hash model* that aids in distinguishing global graph concepts induced by subtrees with similar node distributions or node sets but significantly different structures. Specifically, after we obtain the local clusters, we feed to each local graph concept to a hash model $\Omega(\cdot)$ that returns the graph embedding of it. Then, we concatenate the hashed graph embedding of the local concept to its original embedding to obtain the updated embedding for it. Let $\boldsymbol{m}_o$ be the original embedding of local concept $o$, then the updated embedding $\hat{\boldsymbol{m}}_o$ is

$$\hat{\boldsymbol{m}}_o = \text{CONCAT}\left(\boldsymbol{m}_o, \Omega(o)\right). \tag{10}$$

The steps afterwards remain the same as we discussed in Section 4.1. A hash model should be able to distinguish graph concepts that have the same node distribution or the same node set. For example, the WL-test can be used as a hash model.

## B LEMMAS AND PROOFS

### B.1 REVIEW OF THE WL ALGORITHM

---

**Algorithm 1** The 1-dimensional Weisfeiler-Lehman Algorithm

---

1: **Input:** Graph $G = (\mathcal{V}, \mathcal{E})$, the number of iterations $T$
2: **Output:** Color mapping $\mathcal{X}_G : \mathcal{V} \to \mathcal{C}$
3: **Initialize:** $\mathcal{X}_G(v) := \text{hash}(G[v])$ for all $v \in \mathcal{V}$
4: **for** $t \leftarrow 1$ **to** $T$ **do**
5:    **for each** $v \in \mathcal{V}$ **do**
6:       $\mathcal{X}_G^t(v) := \text{hash}\left(\mathcal{X}_G^{t-1}(v), \{\{\mathcal{X}_G^{t-1}(u) : u \in \mathcal{N}_G(v)\}\}\right)$
7:    **break upon convergence**
8: **Return:** $\mathcal{X}_G^T$

---

### B.2 PROOF OF THEOREM 4.2

*Proof.* We proof Theorem 4.2 by Mathmetical Induction. In the base step, we aim to prove Lemma B.1. In the Inductive step, we aim to prove Lemma B.2.

**Lemma B.1** (Base step). *Given a graph $G = (V, E)$ with the countable input node features $\mathbf{X}$, and a $L$-layer GNN $f(\cdot)$ that updates the layer-wise node-embeddings by Equation (1). Then $\forall v \in V$, the first layer node embedding $\boldsymbol{h}_v^{(1)}$ is a Perfect Rooted Tree Representation of the 1-hop subtree rooted at $v$, if the functions AGG and UPDATE in Equation (1) are injective.*

*Proof.* Let $T_v$ be the 1-hop subtree rooted at $v$ in $G$. Assume $\boldsymbol{h}_v^{(1)}$ is not a Perfect Rooted Tree Representation of $T_v$. Then either of the cases should hold:

(i) There exist another 1-hop subtree $T_u$, embedded by $f(\cdot)$ as $\boldsymbol{h}_u^{(1)}$, which is non-isomorphic to $T_v$, but $\boldsymbol{h}_v^{(1)} = \boldsymbol{h}_u^{(1)}$;

(ii) There exist an isomorphic subtree $T_u$, embedded by $f(\cdot)$ as $\boldsymbol{h}_u^{(1)}$, where $\boldsymbol{h}_v^{(1)} \neq \boldsymbol{h}_u^{(1)}$.

According to Equation (1), we can calculate $\boldsymbol{h}_v^{(1)}$ and $\boldsymbol{h}_u^{(1)}$ by:

$$\boldsymbol{h}_v^{(1)} = \text{UPDATE}^{(1)} \left( \mathbf{X}_v, \text{AGG}^{(1)} \left( \{\mathbf{X}_w : w \in \mathcal{N}(v)\} \right) \right),$$

$$\boldsymbol{h}_u^{(1)} = \text{UPDATE}^{(1)} \left( \mathbf{X}_u, \text{AGG}^{(1)} \left( \{\mathbf{X}_w : w \in \mathcal{N}(u)\} \right) \right),$$

We firstly consider Case (i). If $T_v$ and $T_u$ are non-isomorphic 1-hop subtrees, then $\mathbf{X}_u \neq \mathbf{X}_v$, or the multisets $\{\mathbf{X}_w : w \in \mathcal{N}(v)\} \neq \{\mathbf{X}_w : w \in \mathcal{N}(u)\}$. Recall that an injective function $g(\cdot)$ refers to a function that that maps distinct elements of its domain to distinct elements. That is, $x_1 \neq x_2$ implies $g(x_1) \neq g(x_2)$; $x_1 = x_2$ implies $g(x_1) = g(x_2)$. If $\mathbf{X}_u \neq \mathbf{X}_v$ or $\{\mathbf{X}_w : w \in \mathcal{N}(v)\} \neq \{\mathbf{X}_w : w \in \mathcal{N}(u)\}$, since AGG and UPDATE are injective, we have $\boldsymbol{h}_v^{(1)} \neq \boldsymbol{h}_u^{(1)}$. Hence we have reached a contradiction.

Next, we consider Case (ii). If $T_u$ is isomorphic to $T_v$, then $\mathbf{X}_u = \mathbf{X}_v$ and the multisets $\{\mathbf{X}_w : w \in \mathcal{N}(v)\} = \{\mathbf{X}_w : w \in \mathcal{N}(u)\}$. Since AGG and UPDATE are both injective, we have $\boldsymbol{h}_v^{(1)} = \boldsymbol{h}_u^{(1)}$. Hence we have reached a contradiction.

Therefore, if the functions AGG and UPDATE in Equation (1) are injective, $\boldsymbol{h}_v^{(1)}$ is a Perfect Rooted Tree Representation of the 1-hop subtree rooted at $v$. $\qquad\square$

**Lemma B.2** (Inductive step). *Given a graph $G = (V, E)$, assume the countable node representation $\boldsymbol{h}_v^{(l-1)}$ for $v \in V$ be the Perfect Rooted Tree Representation of the corresponding $(l-1)$-hop subtrees rooted at $v$. We calculate the $l$-th layer representation of $v$, i.e., $\boldsymbol{h}_v^{(l)}$, using Equation (1). Then $\boldsymbol{h}_v^{(l)}$ is a Perfect Rooted Tree Representation of the full $l$-hop subtree rooted at $v$ if the functions AGG and UPDATE are injective.*

*Proof.* Let $T_v^{(l)}$ be the full $l$-hop subtree rooted at $v$ in $G$. Assume $\boldsymbol{h}_v^{(l)}$ is not a Perfect Rooted Tree Representation of the full $l$-hop subtree rooted at $v$. Then, either of the following cases should hold:

(i) There exist another full $l$-hop subtree $T_u^{(l)}$, embedded by $f(\cdot)$ as $\boldsymbol{h}_u^{(l)}$, which is non-isomorphic to $T_v^{(l)}$, but $\boldsymbol{h}_v^{(l)} = \boldsymbol{h}_u^{(l)}$;

(ii) There exist an isomorphic subtree $T_u^{(l)}$, embedded by $f(\cdot)$ as $\boldsymbol{h}_u^{(l)}$, where $\boldsymbol{h}_v^{(l)} \neq \boldsymbol{h}_u^{(l)}$.

According to Equation (1), we can calculate $\boldsymbol{h}_v^{(1)}$ and $\boldsymbol{h}_u^{(1)}$ by:

$$\boldsymbol{h}_v^{(l)} = \text{UPDATE}^{(l)} \left( \boldsymbol{h}_v^{(l-1)}, \text{AGG}^{(l)} \left( \left\{ \boldsymbol{h}_w^{(l-1)} : w \in \mathcal{N}(v) \right\} \right) \right),$$

$$\boldsymbol{h}_u^{(l)} = \text{UPDATE}^{(l)} \left( \boldsymbol{h}_u^{(l-1)}, \text{AGG}^{(l)} \left( \left\{ \boldsymbol{h}_w^{(l-1)} : w \in \mathcal{N}(u) \right\} \right) \right).$$

First, we consider Case (i). If $T_v^{(l)}$ and $T_u^{(l)}$ are non-isomorphic full $l$-hop subtrees, then the $(l-1)$-hop subtrees $T_v^{(l-1)}$ and $T_u^{(l-1)}$ are non-isomorphic, or the multisets $\left\{ \boldsymbol{h}_w^{(l-1)} : w \in \mathcal{N}(v) \right\} \neq \left\{ \boldsymbol{h}_w^{(l-1)} : w \in \mathcal{N}(u) \right\}$.

Since $\boldsymbol{h}_v^{(l-1)}$ is the Perfect Rooted Tree Representation of the corresponding $(l-1)$-hop subtrees rooted at $v$, we have: If $T_v^{(l-1)}$ and $T_u^{(l-1)}$ are non-isomorphic, then $\boldsymbol{h}_v^{(l-1)} \neq \boldsymbol{h}_u^{(l-1)}$. Since the functions AGG and UPDATE are injective, we have $\boldsymbol{h}_v^{(l)} \neq \boldsymbol{h}_u^{(l)}$. Hence we have reached a contradiction.

Next, we consider Case (ii). If $T_u^{(l)}$ is isomorphic to $T_v^{(l)}$, then the $(l-1)$-hop subtress $T_u^{(l-1)}$ and $T_v^{(l-1)}$ are also isomorphic. And we have and the multisets $\left\{ \boldsymbol{h}_w^{(l-1)} : w \in \mathcal{N}(v) \right\} = \left\{ \boldsymbol{h}_w^{(l-1)} : w \in \mathcal{N}(u) \right\}$. Since $\boldsymbol{h}_v^{(l-1)}$ is the Perfect Rooted Tree Representation of the corresponding $(l-1)$-hop subtrees rooted at $v$, we have $\boldsymbol{h}_v^{(l-1)} = \boldsymbol{h}_u^{(l-1)}$. Since AGG and UPDATE are both injective, we have $\boldsymbol{h}_v^{(l)} = \boldsymbol{h}_u^{(l)}$. Hence we have reached a contradiction.

Therefore, if the functions AGG and UPDATE in Equation (1) are injective, $\boldsymbol{h}_v^{(l-1)}$ for $v \in V$ be the Perfect Rooted Tree Representation of the corresponding $(l-1)$-hop subtrees rooted at $v$, then $\boldsymbol{h}_v^{(l)}$ is a Perfect Rooted Tree Representation of the 1-hop subtree rooted at $v$. □

The following lemma shows that if the input of a GNN is countable, then the node embeddings are also countable.

**Lemma B.3.** *(Xu et al., 2019) Assume the input feature $\mathcal{X}$ is countable. Let $g^{(l)}$ be the function parameterized by a GNN's l-th layer for $l = 1, \ldots, L$, where $g^{(1)}$ is defined on multisets $X \subset \mathcal{X}$ of bounded size. The range of $g^{(l)}$, i.e., the space of node hidden features $\boldsymbol{h}_v^{(l)}$, is also countable for all $l = 1, \ldots, L$.*

This lemma implies that if the input $\mathbf{X}_v$ for any $v$ is countable, then $\boldsymbol{h}_v^{(l)}$ for any $l$ is also countable, making our assumption in Lemma B.2 valid.

Hence, we have proved Theorem 4.2 using Mathmetical Induction.

□

### B.3 PROOF OF COROLLARY A.1

*Proof.* We first prove (i). Firstly, we assume $\boldsymbol{h}_{v,\text{AGG}}^{(l)}$ is a Perfect Rooted Tree Representation of the full $l$-hop subtree $T_v$ rooted at $v$. By Definition 4.1, for any arbitrary same-depth rooted tree $T_u$ in the same countable space, $\boldsymbol{h}_{v,\text{AGG}}^{(l)} = \boldsymbol{h}_{u,\text{AGG}}^{(l)}$ if and only if $T_v$ is isomorphic to $T_u$. Since UPDATE is injective, we have for any arbitrary same-depth rooted tree $T_u$ in the same countable space, $\boldsymbol{h}_v^{(l)} = \boldsymbol{h}_u^{(l)}$ if and only if $T_v$ is isomorphic to $T_u$. Therefore, if UPDATE is injective and $\boldsymbol{h}_{v,\text{AGG}}^{(l)}$ is a Perfect Rooted Tree Representation of the full $l$-hop subtree $T_v$ rooted at $v$, then $\boldsymbol{h}_v^{(l)}$ is a Perfect Rooted Tree Representation of the full $l$-hop subtree rooted at $v$.

Secondly, we assume $\boldsymbol{h}_{v,\text{AGG}}^{(l)}$ is not a Perfect Rooted Tree Representation of the full $l$-hop subtree $T_v$ rooted at $v$. Then Definition 4.1 does not hold, which means either of the following cases holds:

- $T_v$ and $T_u$ are isomorphic, but $\boldsymbol{h}_{v,\text{AGG}}^{(l)} \neq \boldsymbol{h}_{u,\text{AGG}}^{(l)}$;

- $T_v$ and $T_u$ are non-isomorphic, but $\boldsymbol{h}_{v,\text{AGG}}^{(l)} = \boldsymbol{h}_{u,\text{AGG}}^{(l)}$.

Since UPDATE is injective, either of the following cases holds:

- $T_v$ and $T_u$ are isomorphic, but $\boldsymbol{h}_v^{(l)} \neq \boldsymbol{h}_u^{(l)}$;

- $T_v$ and $T_u$ are non-isomorphic, but $\boldsymbol{h}_v^{(l)} = \boldsymbol{h}_u^{(l)}$.

Therefore, if UPDATE is injective and $h_{v,\text{AGG}}^{(l)}$ is not a Perfect Rooted Tree Representation of the full $l$-hop subtree $T_v$ rooted at $v$, then $h_v^{(l)}$ is not a Perfect Rooted Tree Representation of the full $l$-hop subtree rooted at $v$.

Hence we can conclude that if UPDATE is injective, then $h_v^{(l)}$ is a Perfect Rooted Tree Representation of the full $l$-hop subtree rooted at $v$ if and only if $h_{v,\text{AGG}}^{(l)}$ is a Perfect Rooted Tree Representation of the full $l$-hop subtree rooted at $v$.

Similarly, we prove (ii). Firstly, we assume $h_{v,\text{AGG}}^{(l)}$ is a mapping of both a full $l$-hop subtree $T_v^{(l)} = (V_{T_v^{(l)}}, E_{T_v^{(l)}})$ and another non-isomorphic $l$-hop subtree $T_v^{(l)\prime} = (V_{T_v^{(l)\prime}}, E_{T_v^{(l)\prime}})$ rooted at $v$. Let $h_{v,\text{AGG}}^{(l)}$ denote the intermediate $l$-th layer embedding derived by the AGG function on $T_v^{(l)}$, $h_{v,\text{AGG}}^{(l)\prime}$ denote the intermediate $l$-th layer embedding derived by the AGG function on $T_v^{(l)\prime}$. Then we get $h_{v,\text{AGG}}^{(l)} = h_{v,\text{AGG}}^{(l)\prime}$. Since UPDATE is injective, $h_v^{(l)} = h_v^{(l)\prime}$, where $h_v^{(l)\prime}$ is the node embedding computed using the same function as $h_v^{(l)}$, but on $T_v^{(l)\prime}$. Therefore, if UPDATE is injective and $h_{v,\text{AGG}}^{(l)}$ is a mapping of both a full $l$-hop subtree and another $l$-hop subtree rooted at $v$, then $h_v^{(l)}$ is a mapping of the same pair of non-isomorphic trees rooted at $v$.

Finally we assume $h_{v,\text{AGG}}^{(l)}$ is not a mapping of both a full $l$-hop subtree and another $l$-hop subtree rooted at $v$. In other words, $T_v^{(l)}$ and a non-isomorphic subtree $T_v^{(l)\prime}$ always have their distinct representations $h_{v,\text{AGG}}^{(l)}$ and $h_{v,\text{AGG}}^{(l)\prime}$, where $h_{v,\text{AGG}}^{(l)} \neq h_{v,\text{AGG}}^{(l)\prime}$. Since UPDATE is injective, we have $h_v^{(l)} \neq h_v^{(l)\prime}$. Therefore, if UPDATE is injective and $h_{v,\text{AGG}}^{(l)}$ is not a mapping of both a full $l$-hop subtree and another $l$-hop subtree rooted at $v$, then $h_v^{(l)}$ is a mapping of both a full $l$-hop subtree and another non-isomorphic $l$-hop subtree rooted at $v$ at the same time.

Hence we conclude that if UPDATE is injective, then $h_v^{(l)}$ is a mapping of both a full $l$-hop subtree and another non-isomorphic $l$-hop subtree rooted at $v$, *if and only if* $h_v^{(l)}$ is a mapping of both a full $l$-hop subtree and another non-isomorphic $l$-hop subtree rooted at $v$ at the same time.

$\square$

## C  STATISTICS OF DATASETS AND MODELS

Dataset Statistics are presented in Table 5. Implementation details of the GNNs for explainablity in this paper is shown in Table 6. The GNNs were trained and evaluated by randomly splitting the datasets into training/validation/testing sets at 0.8/0.1/0.1 ratio. The random seed we used was 1234 while we split the data. For the baseline performance reported in Table 1 and 2, we utilize their implementations in the DGL library (Wang et al., 2019) and evaluate their performance at sparsity=0.5. All the experiments were conducted on a machine with an Intel Core i7-10700K processor with 64 GB RAM and a single NVIDIA GeForce RTX 3090 GPU.

Table 5: Statistics of datasets.

| Datasets | BA-2Motifs | | BAMultiShapes | | Mutagenicity | | NCI1 | |
|---|---|---|---|---|---|---|---|---|
| | #nodes | #edges | #nodes | #edges | #nodes | #edges | #nodes | #edges |
| mean | 25 | 51 | 40 | 87.5 | 30.3 | 61.5 | 29.9 | 64.6 |
| std | 0 | 1 | 0 | 7.2 | 20.1 | 33.6 | 13.6 | 29.9 |
| min | 25 | 49 | 40 | 78 | 4 | 6 | 3 | 4 |
| quantile25 | 25 | 50 | 40 | 78 | 19 | 38 | 21 | 46 |
| median | 25 | 50 | 40 | 90 | 27 | 56 | 27 | 58 |
| quantile75 | 25 | 52 | 40 | 92 | 35 | 76 | 35 | 76 |
| max | 25 | 52 | 40 | 100 | 417 | 224 | 111 | 238 |
| #graphs | 1000 | | 1000 | | 4337 | | 4110 | |

Table 6: Details of the GNN models used to produce our experimental results, where "hidden" is the latent dimension size, and L is the number of GNN layers.

| Datasets | BA-2Motifs | BAMultiShapes | Mutagenicity | NCI1 |
|---|---|---|---|---|
| number of GNN layers | 3 | 3 | 3 | 3 |
| hidden | 32 | 32 | 64 | 64 |
| global pooling | mean | mean | mean | mean |
| layer type | GIN | GIN | GIN | GIN |
| learning rate | 0.01 | 0.01 | 0.01 | 0.01 |
| batch size | 256 | 256 | 256 | 256 |
| epochs | 200 | 200 | 200 | 200 |
| train acc | 1.00 | 0.99 | 0.91 | 0.95 |
| test acc | 1.00 | 0.97 | 0.81 | 0.80 |

# D    MORE EXPERIMENTAL RESULTS

## D.1    ADDITIONAL RESULTS ON BAMULTISHAPES AND NCI1

Figure 5 presents the global explanations produced by various global explainers on BAMultiShapes and NCI1 datasets. For the NCI1 dataset, we cannot map the node type numbers to the actual atoms because that information was not available. So, the explanations we provide only include the node type numbers. GLGExplainer generates long Boolean formulas on BAMultiShapes dataset. However, it fails to identify the house motif. Moreover, the predicates in the Class 1 formula, namely $(P_1 \wedge P_3)$, $(P_2 \wedge P_5)$ and $(P_5 \wedge P_1)$, are not faithful to the ground-truth, as they require the presence of multiple grids or multiple wheels in Class 1 graphs. Recall that the ground-truth rules of BAMultiShapes is that Class 0 includes plain BA graphs and those with individual motifs or a combination of all three, whereas Class 1 comprises BA graphs enriched with any two of the three motifs. Furthermore, the Boolean formulas from GLGExplainer on NCI1 is a bit confusing, since $P_0 \vee (P_0 \wedge P_1)$ is logically equivalent to $P_0$. Consequently, for the NCI1 dataset, GLGExplainer only provides random local explanations of each prototype, where Prototype 0 stands for Class 0 and Prototype 1 stands for Class 1. The insights provided by these random local explanations are less informative. On the other hand, the global explanations from GCNeuron are relatively less intuitive as negations are frequently involved, and they are challenging for humans to understand.

In contrast, our approach successfully identifies all the outstanding motifs for the BAMultiShapes dataset, namely, the house, wheel, as well as grid motifs. In particular, TreeX recognize the patterns in Barabasi-Albert (BA) graphs as the Class 0 motifs, and house, wheel, grid as the Class 1 motifs, by identifying higher confidence on these motifs at each class. This is reasonable because, as shown in Table 5, all the data samples in BAMultiShapes contain 40 nodes. Hence, if more house, wheel or grid motifs are included in a graph, then less BA patterns will be in it. Given that all the Class 1 graphs contain two out of three motifs in house, wheel or grid, whereas most Class 0 graphs contain at most one of the three motifs, it is reasonable for the GNNs to consider that the Class 0 graphs contains a larger portion of BA patterns than the Class 1 graphs. Additionally, neither TreeX nor GLGExplainer is able to capture the ground truth rule that graphs contain all of the three motifs are Class 0 graphs, which is as expected, since as shown in Table 6, the GIN does not achieve perfect accuracy on BAMultiShapes. These experimental results have demonstrate that our approach has the potential to provide insights into some occasionally incorrect rules learned by the model. For the NCI1 dataset, the inherent design of TreeX allows it to capture larger graph patterns than GLGExplainer and GCNeuron.

## D.2    HYPERPARAMETERS IN TREEX

TreeX involves several hyperparameters, including the number of local clusters $k$ in local subtree mining, the number of global clusters $m$ in global clustering, and the weighting factor $\lambda$ while generating global rules. In this section, we present the hyperparameter settings used in our experiments and explain how we determine them.

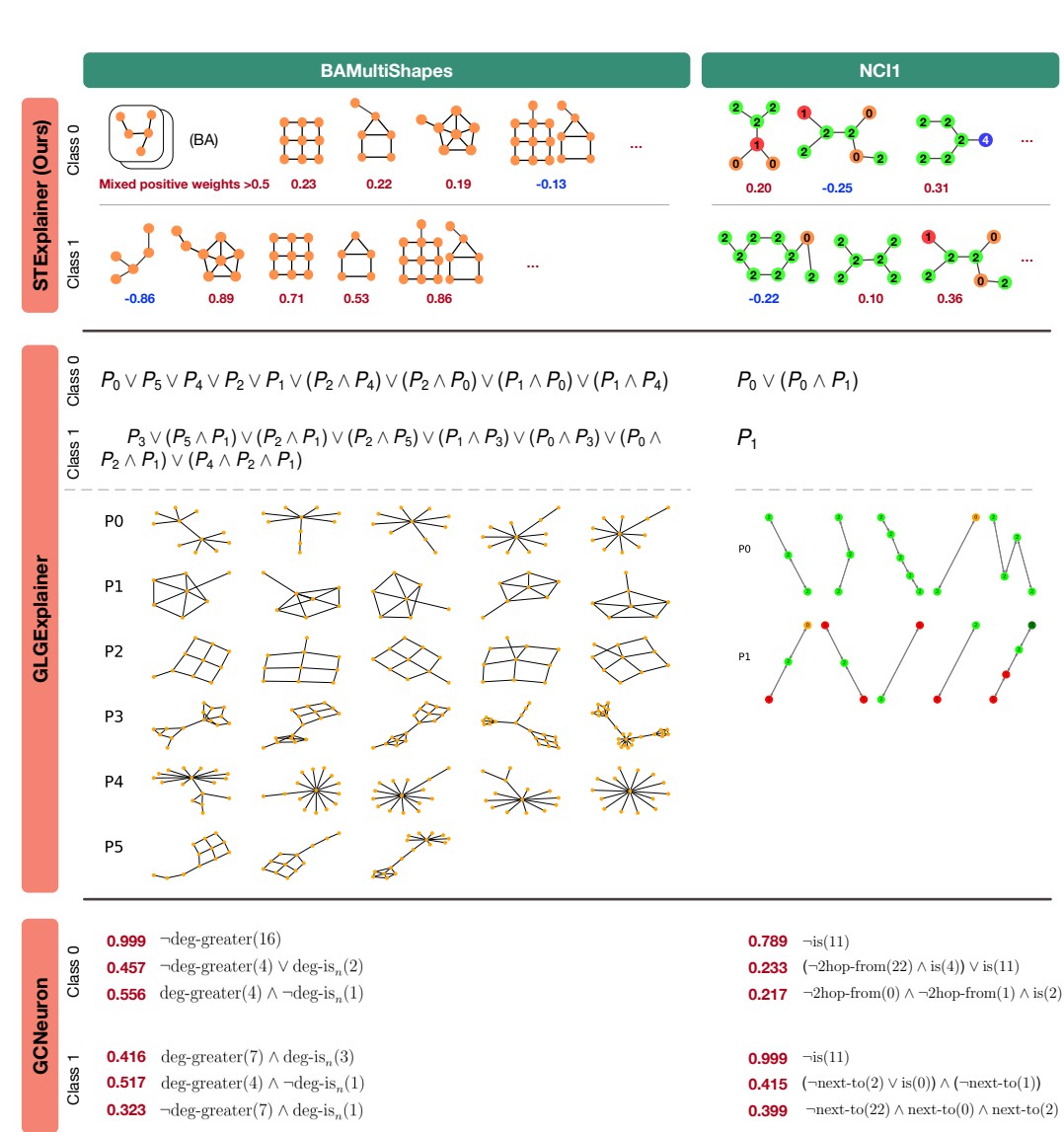

Figure 5: Global explanations by TreeX (ours), GCNeuron and GLGExplainer on BAMultiShapes and NCI1 datasets. We run both baseline methods so that they explain the same GNN models as our approach.

**Number of local clusters** $k$. The choice of this hyperparameter has minimal impact on the results of the global-level explanation, as long as $k$ is not excessively small. For instance, when we select $k$ from the range $3, 4, 5, 6, 7, 8, 9, 10$, TreeX consistently identifies the five-node cycle and the house motifs with high confidence. This is because the local clustering algorithm, whether $k$-means or the EM algorithm, automatically adapts by shrinking to a smaller number of clusters when a larger number is allowed. Generally, we recommend setting $k$ to be 1 to 3, plus the number of classes. In our experiments, we set $k = 3$ for BA-2Motifs and $k = 5$ for all other datasets.

**Number of global clusters** $m$. We determine the number of global concepts with the help of the prediction accuracy fidelity performance. We plot the accuracy fidelity performance with respect to the number of global concepts in Figure 6. It shows that TreeX can achieve high fidelity performance even with a small number of global concepts.

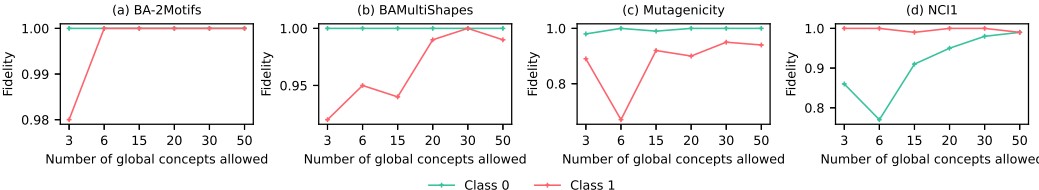

Figure 6: Fidelity performance of TreeX, with respect to the number of global concepts, using Kmeans as the local clustering algorithm.

To prevent the number of global clusters from being too small, we evaluate the fidelity across various values of $m$ and choose the one that achieves nearly optimal fidelity performance. In our experiments, we set $m = 6, 30, 30, 30$ for the four datasets repectively. It's important to note that since $m$ also determines the dimension of the trainable parameter $\boldsymbol{w}$, setting this value to be excessively large may result in increased training time and is not recommended.

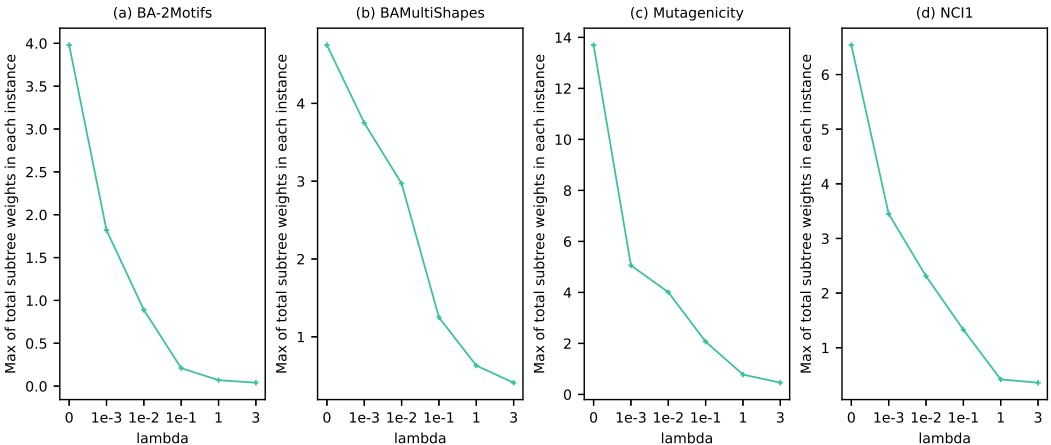

Figure 7: Maximum of total subtree weights in each instance with respect to $\lambda$ on each dataset.

**Weighting factor** $\lambda$. As we discussed in Section 4.1, we introduce a penalty term on the trainable weights $\boldsymbol{w}$ to emulate global-level pooling commonly utilized in GNNs. For instance, in the case of GNNs employing global mean pooling, we regulate $\lambda$ to ensure that $\sum(\boldsymbol{w}\mathbf{K}_i)$ for each instance $G_i$ does not exceed 1. It's important to note that $\sum(\boldsymbol{w}\mathbf{K}_i)$ is not strictly constrained to equal 1 because some unimportant features may not necessarily contribute to increasing the likelihood of predicting the target class. And if the sum significantly exceeds 1, it may lead to unexpected behavior, as the pooled embedding may fall outside the expected distribution. We aim to control $\lambda$ so that $\max([\sum(\boldsymbol{w}\mathbf{K}_i), \forall G_i \in \mathcal{D}])$ is within the range of $[0.1, 1]$. Figure 7 shows the maximum of total subtree weights in each instance with respect to $\lambda$ on each dataset. Based on the results, we choose $\lambda = 0.01$ for BA-2Motifs, $\lambda = 1$ for BAMultiShapes, Mutagenicity and NCI1.

### D.3 VISUALIZATION OF GLOBAL EXPLANATIONS ON DATA INSTANCES

Figure 8, 9, 10, 11 visualize the global explanations extracted using our approach on the actual data instances. We can easily observe that the five-node cycles and house motifs are accurately highlighted on the graphs at the corresponding classes in the BA-2Motifs dataset. For BAMultiShapes, the BA patterns for Class 0, as well as house, wheel, grid motifs for Class 1 are clearly illustrated. On Mutagenicity and NCI1, TreeX is able to highlight functional groups such as "-NO2", "-NH2", "-NO".

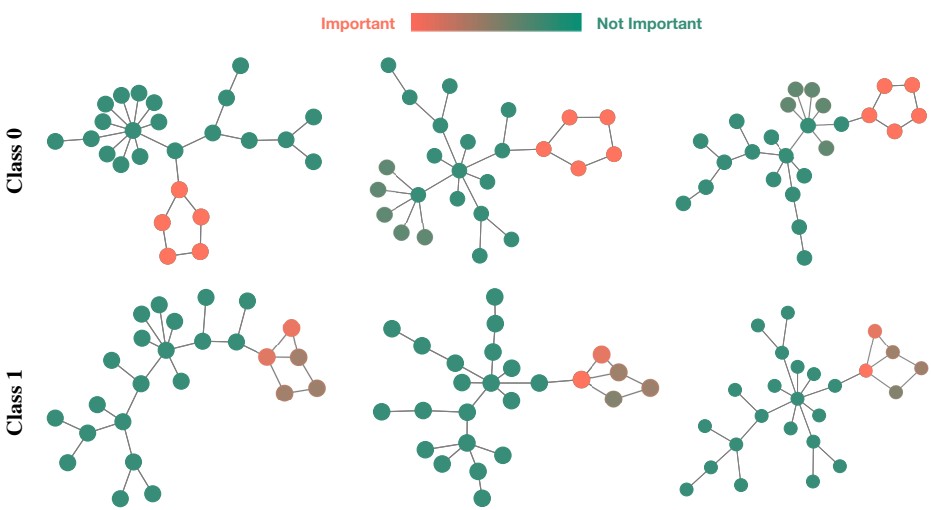

Figure 8: Visualization of the global explanations extracted by TreeX on the actual data instances of the BA-2Motifs dataset.

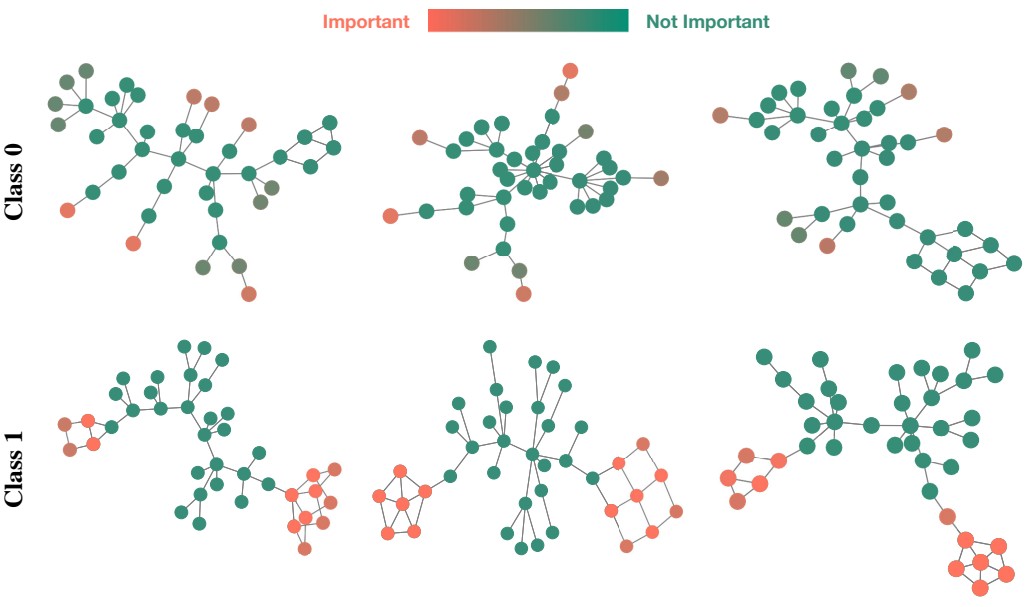

Figure 9: Visualization of the global explanations extracted by TreeX on the actual data instances of the BAMultiShapes dataset.

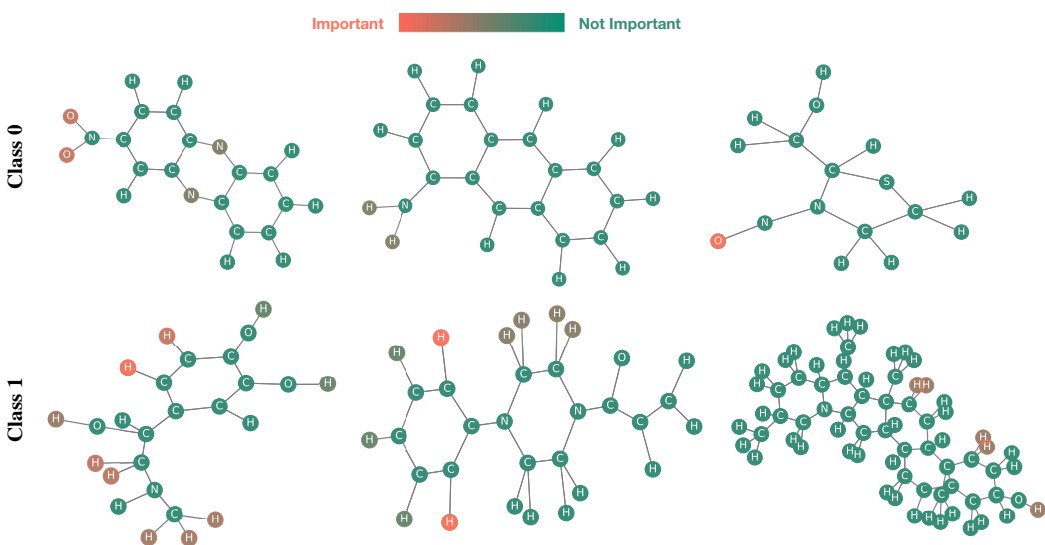

Figure 10: Visualization of the global explanations extracted by TreeX on the actual data instances of the Mutagenicity dataset.

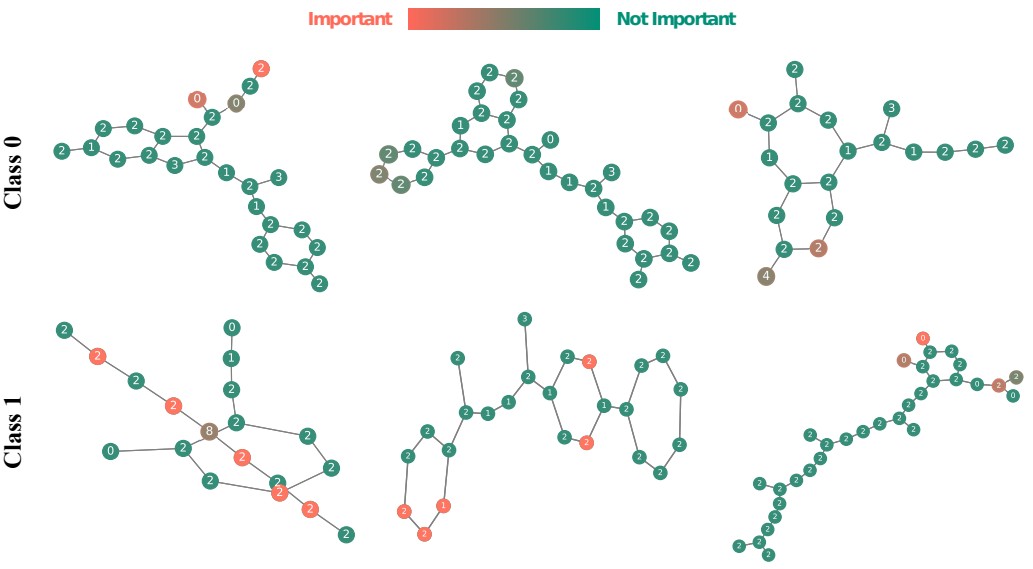

Figure 11: Visualization of the global explanations extracted by TreeX on the actual data instances of the NCI1 dataset.

