# OpenReview forum: "TreeX: Generating Global Graphical GNN Explanations via Critical Subtree Extraction"
_ICLR.cc/2025/Conference — Submitted to ICLR 2025_

### Official Review · Reviewer_E6Gj · 2024-10-31

**Soundness:** 2
**Presentation:** 3
**Contribution:** 2
**Rating:** 3
**Confidence:** 4

**Summary:**

This paper proposed a method to explain message-passing GNNs via critical subtree extraction. It analyzes and extracts critical subtrees incurred by the inner workings of message passing. An efficient algorithm which doesn't require  complex subgraph search is proposed to aggregate subtrees in the embedding space. As a result, we can make intuitive graphical explanations for Message-Passing GNNs on local, class and global levels.

**Strengths:**

The paper made two innovative contributions:
1. It providing a global explanations of GNNs on the dataset or class level in the format of subgraphs rather than nodes, language rules or prototype embeddings in the previous literature.
2. Instead of subgraph enumeration or search, the paper proposed an efficient algorithm for subtree extraction, and using the root node embedding as subtree embedding.

**Weaknesses:**

I have concern on the global concept extraction method outlined between lines 200-206. How can the method ensure that the spaces of embedding across $D$ graph instances are aligned? The authors should provide a detailed explanation of this process, as without it, I am skeptical about the possibility of clustering the $kD$ local graph concepts, because these concepts originate from $D$ distinct embedding spaces.

In the "global rule generation for each class" (line 220-), I doubt the method can always work in general situations. The global rules are generated through frequency-based analysis, and it may be challenging to ensure that these frequency-based features always possess the necessary discriminative power to distinguish the differences between classes.

The innovation of this paper is not significant. Please compare the proposed method with
- Motif-driven Subgraph Structure Learning for Graph Classification (https://arxiv.org/abs/2406.08897)
- STExplainer: Global Explainability of GNNs via Frequent SubTree Mining (https://openreview.net/forum?id=HgSfV6sGIn)

**Questions:**

P196: Should the $max$  operator or $top_k$ operator be in equation 4?
P264:  presentaions --> representations?

---

> ### Author Response · Authors · 2024-12-02
>
> > Q: In the global concept extraction method, I am skeptical about the possibility of clustering the kD local graph concepts, because these concepts originate from D distinct embedding spaces.
>
> **A:** Your concern about the alignment of embedding spaces across the $D$ graph instances is addressed by the nature of our method. The concepts are not derived from $D$ distinct embedding spaces. Instead, our work focuses on post-hoc GNN explainability, where the parameters of the GNN are fixed during the extraction of explanation concepts across the entire dataset. This ensures that all node embeddings from all graph instances reside within the same embedding space. In this shared space, closer embeddings inherently represent similar subgraph patterns, enabling effective clustering of the $kD$ local graph concepts without misalignment issues.
>
> > Q:  The global rules are generated through frequency-based analysis, and it may be challenging to ensure that these frequency-based features always possess the necessary discriminative power to distinguish the differences between classes.
>
> **A:** We train a linear mapping from the most significant global concepts—which can be as numerous as needed, depending on the specific task—to maximize the prediction probability for each class, effectively forming a global rule for each class.
> Since the original GNN is trained to distinguish between classes, its embeddings inherently capture class-specific information. By mapping these concepts to class labels, we ensure they possess the necessary discriminative power, as long as the original GNN effectively separates the classes. This combination of frequency-based concept selection and supervised learning ensures that our global rules can reliably distinguish differences between classes.
>
> > Q: Comparing with a concurrent work and a rejected work.
>
> **A:** Regarding the concurrent work recently uploaded to ArXiv, we will certainly discuss it in our paper. It is a motif-driven Graph Structure Learning approach aimed at improving graph classification. While it focuses more on developing a self-explainable graph learning model, our work emphasizes post-hoc explainability. As for the rejected work, we believe it is fair to allow the authors the opportunity to reuse and refine their ideas instead of treating it a comparable work.
>
> > Q: Should the max operator or topk operator be in equation 4?
>
> **A:** It should be the max operator, as we select only the most frequently overlapped edges in our experiments. However, the top-k operator can also be considered by setting k to 1 or more.

---

### Official Review · Reviewer_1wwV · 2024-11-01

**Soundness:** 2
**Presentation:** 2
**Contribution:** 3
**Rating:** 3
**Confidence:** 4

**Summary:**

TreeX introduces an explanation method primarily for graph classifiers, providing both local and global explanations based on the subgraph concept derived from their proposed subtree extraction method. Local concepts are extracted from subtrees in individual instance graphs. Final local concepts for each input graph are obtained by clustering through $k$-means. Global concepts are then extracted by aggregating and clustering these local concepts. To understand the impact of the discovered global concepts, the weight of each concept is optimized to represent the prediction of a certain class.

**Strengths:**

Providing local and global explanations is crucial for understanding predictive outputs at both individual and class levels. Figure 1 offers an insightful perspective on how each instance's prediction compares to class-specific patterns, making it easier to debug predictive outputs. Furthermore, extracting concepts based on subtrees presents a novel approach to mining meaningful patterns.

**Weaknesses:**

W1. The proposed method's heavy reliance on K-means clustering for both local and global concept extraction introduces several inherent weaknesses. Consequently, the proposed method exposes the weaknesses and limitations of clustering such as initialization issues, sub-optimal issues, the sensitivity of hyper-parameter $k$, and quality issues such as under-clustering or over-clustering. Due to this well-known limitation, the assignment of clusters is likely to be sensitive. Thus, explanations based on the clustering method may not be robust.

W2. The process of providing final explanations in Figures 8, 9, 10, and 11 is not clear. Based on class-specific global rules,  connecting concepts and subgraphs in the specific input graph.

W3. Experiments are not thorough in several aspects as below.

(1)  A quantitative evaluation of global explanations is lacking, which is a key contribution of the proposed method. This aspect should be more thoroughly examined.

(2) Global explainers in the graph domain such as XGNN [1] and GNNInterpreter [2] are not considered in the comparison.

(3) No visualization of the local explanations.

(4) AccFidelity is an easy metric to achieve decent performance as it counts the explanation as correct even when $\hat{y}_i=0.51$ in binary classification tasks. Nevertheless, authors only use $Fid^{-}$, excluding the $Fid^{+}$ [3] which is the most common measurement in many literatures.

(5) The time analysis focuses solely on explanation inference, omitting the process of concept extraction. This process likely incurs significant computational costs due to its use of k-means algorithms in twice and post-processing steps to extract final concepts. A more comprehensive time analysis would provide a clearer picture of the method's efficiency.

Minor typo in line 352, e.g., In the he BA-2Motifs data.

Reference

[1] Yuan, Hao, et al. "Xgnn: Towards model-level explanations of graph neural networks." *Proceedings of the 26th ACM SIGKDD international conference on knowledge discovery & data mining*. 2020.

[2] Wang, Xiaoqi, and Han-Wei Shen. "Gnninterpreter: A probabilistic generative model-level explanation for graph neural networks." *arXiv preprint arXiv:2209.07924* (2022).

[3] Yuan, Hao, et al. "Explainability in graph neural networks: A taxonomic survey." *IEEE transactions on pattern analysis and machine intelligence* 45.5 (2022): 5782-5799.

**Questions:**

Q1. The authors currently evaluate the performance of the proposed method under different random seeds when splitting the data. Given that clustering algorithms are sensitive to initialization based on random seeds, how do the explanation results change under different cluster assignments resulting from different initialization?

Q2. Why are the extraction processes for local concepts in Eq. 4 and global concepts (which are closest to the center of cluster $U_j$) from clusters different? Can you further discuss the motivation behind the proposed method?

Q3. How can explicit local explanations be derived from the global rules in the form of presenting important subgraphs?

---

> ### Author Response · Authors · 2024-12-02
>
> We are committed to clarify any misunderstandings by the reviewer. Please let us know if you have any further questions or concerns.
>
> > **Q:** The explanations based on the clustering method may not be robust since it may be sensitive to hyperparameter $k$.
>
> **A:** We address the concern regarding sensitivity to the hyperparameter k through a comprehensive hyperparameter analysis provided in Appendix D.2. In our experiments, we used random initialization for clustering and tested various values for k . The performance of our method remained consistent, achieving consistently high fidelity results. This demonstrates that our method is robust to changes in the number of clusters, further validating the stability and reliability of our overall design.
>
> > **Q (W2 and Q3):** The process of providing final explanations in Figures 8, 9, 10, and 11 is not clear. Based on class-specific global rules, connecting concepts and subgraphs in the specific input graph.
>
> **A:** The explanations in Figures 8, 9, 10, and 11 follow the method outlined in Section 4.3. For each instance, we extract local subgraph concepts using Local Concept Mining Based on Subtrees (Section 4.1) and align them with the nearest global concepts. A concept mask $K$ is constructed to represent the presence of these concepts, and their importance for the target class $y_t$ is calculated using the trained weights $w_t $ from the global rules ($ I_t = K w_t $). This produces instance-specific explanations as a linear combination of global concepts, highlighting their contributions to the class prediction at each graph instance.
>
> > **Q:** A quantitative evaluation of global explanations is lacking.
>
> **A:** We have conducted a thorough quantitative evaluation of the global explanations generated by our method by applying the global rules to each data instance and computing the fidelity, as demonstrated in Table 1 and Table 2. These results clearly show that our approach produces global rules that effectively explain the GNNs.
>
> It is important to note that none of the existing works provide a direct quantitative evaluation of their global explanations—most do not include quantitative comparisons at all. Quantitatively evaluating global explanations is inherently challenging because global rules are often single expressions, making it difficult to measure their validity in terms of percentages or statistical metrics. To address this, we assessed the applicability of the global rules across individual data instances as we just mentioned, allowing us to quantify their effectiveness.
> In addition to the quantitative results, we compared the global rules extracted by our method with those from other approaches in Figure 3 and Figure 5. These comparisons illustrate that our global explanations are more reasonable and accurately capture the critical motifs relevant to the task.
>
> > **Q:** Global explainers in the graph domain such as XGNN [1] and GNNInterpreter [2] are not considered in the comparison.
>
> **A:** We have discussed these works in Lines 103–106, noting that XGNN [1] and GNNInterpreter [2] are generation-based explainers that produce numerous examples for each target class but do not provide clear, interpretable concepts. Instead, they rely on human observers to interpret the results.
> In contrast, our method focuses on factual, global-level explanations, similar to GLG-Explainer (Azzolin et al., 2023) and GCNeuron (Xuanyuan et al., 2023). Since our research targets a different focus than [1] and [2], we chose GLG-Explainer and GCNeuron as our baselines. As shown in Figures 3 and 5, our method, TreeX, effectively generates clear subgraph concepts as global explanations, in contrast to GLG-Explainer and GCNeuron, which produce less interpretable boolean rules between latent clusters or natural language descriptions.
>
> > **Q:** No visualization of the local explanations.
>
> **A:** We included visualizations of the local explanations in Appendix D.3. Our method derives local explanations by first extracting the global explanation and then applying it to each specific instance to generate the corresponding local explanations.

---

> ### Author Response · Authors · 2024-12-02
>
> > Q: AccFidelity is an easy metric to achieve decent performance as it counts the explanation as correct even when y=0.51 in binary classification tasks. Nevertheless, authors only use Fid-, excluding the Fid+ [3] which is the most common measurement in many literatures.
>
> **A:** As discussed in [3], AccFidelity is not a trivial metric but a crucial one. A model might perform well on ProbFidelity while yielding poor results on AccFidelity. For example, consider two instances with original prediction probabilities $ y_1 = 0.51 $ and $ y_2 = 0.52 $. If an explainer generates local explanations with new probabilities $ y_1' = 0.49 $ and $y_2' = 0.99 $, the ProbFidelity, calculated as $\frac{1}{2} \left( (0.51 - 0.49) + (0.52 - 0.99) \right) = -0.225 $, indicates good performance since lower ProbFidelity is better. However, the AccFidelity, calculated as $ \frac{1}{2} \left( 0 + 1 \right) = 0.5 $, is significantly worse than the ideal score of 1.0, as higher AccFidelity is better. This example highlights the importance of considering both metrics to comprehensively evaluate the performance of an explainer. In our paper, we assess our method on both AccFidelity and ProbFidelity, demonstrating strong performance on both metrics.
>
> We do not assess Fid+ for our method because it is not applicable to TreeX. Fid+ is designed for explainers that identify important features and evaluate the change in prediction probability by excluding those critical features. However, TreeX goes beyond simply identifying important subgraph concepts—it assigns an importance coefficient to each concept and constructs a linear combination of these concepts. These coefficients are integral to the explanation and are used to calculate the prediction probability. Due to this design, it is not feasible to exclude the explanation subgraphs to compute Fid+.
> It is important to note that evaluation metrics are meant to validate the performance of the explainer. We have evaluated TreeX using appropriate metrics, specifically AccFidelity and ProbFidelity, and the results demonstrate its superior performance compared to existing methods.
>
> > Q: The time analysis focuses solely on explanation inference, omitting the process of concept extraction.
>
> **A:** The time analysis does not focus solely on explanation inference. As stated in the caption of Table 4, the reported results reflect the elapsed time from the very beginning of the entire process, including concept extraction and $k$-means clustering. These results demonstrate that our method is more efficient than even local-level explanation methods.
>
> > Q: The authors currently evaluate the performance of the proposed method under different random seeds when splitting the data. Given that clustering algorithms are sensitive to initialization based on random seeds, how do the explanation results change under different cluster assignments resulting from different initialization?
>
> **A:** As noted in Line 955, we fix the random seed to 1234 in our experiments to ensure consistency and reproducibility, following standard practices in research where random initialization is required. By doing so, we eliminate variability due to random cluster assignments, allowing us to focus on evaluating the performance of our method under controlled and reproducible conditions. While clustering algorithms can be sensitive to initialization, our fixed-seed approach ensures that the explanation results are stable and reliable across repeated experiments.
>
> > Q: Why are the extraction processes for local concepts in Eq. 4 and global concepts (which are closest to the center of cluster Uj) from clusters different? Can you further discuss the motivation behind the proposed method?
>
> **A:** The extraction processes for local and global concepts differ because they serve distinct purposes. In local concept extraction, as shown in Fig. 2, we cluster all subtrees within a single graph instance to identify critical subgraph structures, where each cluster represents a local concept. To ensure clarity, we use the overlapping subgraph structures of the subtrees within each cluster as the representation for the local concept. In contrast, global concept extraction clusters all local concepts across the dataset into global clusters, representing shared patterns across multiple graph instances. To capture the most representative pattern for each global concept, we select the local concept closest to the centroid of its global cluster. This difference is motivated by the need for different levels of abstraction: local concepts emphasize fine-grained patterns within individual graphs, while global concepts generalize patterns across the dataset. This hierarchical approach ensures that both local and global concepts are interpretable and representative of their respective objectives, enabling effective and scalable explanations.

---

### Official Review · Reviewer_Ubir · 2024-11-03

**Soundness:** 3
**Presentation:** 2
**Contribution:** 2
**Rating:** 3
**Confidence:** 3

**Summary:**

The paper presents TreeX, a method for GNN explainability. It extracts subtree information to construct local concepts, global concepts, and further generate rules for class-specific interpretations. TreeX is computationally efficient compared to previous subgraph-based methods, as it leverages message passing to capture subtree features, while still delivering meaningful concepts for interpretability.

**Strengths:**

The proposed method is fast comparing to previous subgraph based method with last layer embedding after message aggregation, which makes it potentially more scalable comparing to subgraph based enumeration method.

The proposed method can cover both local and global concept extraction.

**Weaknesses:**

The subtree features extracted by last layer embedding might not be able to fully reflect meaningful concept, and could ignore certain patterns that are useful for explainability. See questions for more details.
The evaluation is not sufficient, deeper experiments should be done on more architectures (GCN, GraphSAGE, GAN, etc), as different GNN architectures have varying expressive power and aggregation mechanisms. Besides, variations in last layer embedding quality across architectures could influence the interpretability in TreeX.

**Questions:**

I have some questions regarding the methodology for constructing concepts based on the last-layer embeddings.

How does the method ensure these clusters are reflecting meaningful, distinct structural concepts within the graph? Elaborated below.
The paper uses last-layer embeddings to construct concept. It is true that the last layer embedding for a specific root node aggregates the nodes within L-hops. However, this aggregation would overlook finer subgraph features (i.e. specific motifs or relational patterns[1][2]). Will this affect the explainability and how does the method address this limitation?
Additionally, how to validate the meaningfulness of these clusters?  Regarding the clustering algorithm, given the similarities in last-layer embeddings, k-means may face issues that embeddings can be smooth or overlapping. As a result, it might not distinguish nuanced structural variations effectively. Did you evaluate whether k-means clustering accurately captures meaningful and distinct concepts in the last-layer embeddings? Do you consider alternative clustering approaches for finer clustering?
[1] GNNExplainer: Generating Explanations for Graph Neural Networks
[2] PGM-Explainer: Probabilistic Graphical Model Explanations for Graph Neural Networks

---

> ### Author Response · Authors · 2024-12-02
>
> We summarize the reviewer's concerns as follows:
>
> i) Subtrees may not sufficiently explain GNNs compared to subgraphs or motifs.
>
> ii) K-means might not effectively distinguish nuanced structural variations.
>
> iii) Discussion on other variants of GNNs.
>
> > **Q:** The subtree features extracted by the last layer's embeddings might not fully reflect meaningful concepts, potentially overlooking finer subgraph features (e.g., specific motifs or relational patterns [1][2]). Will this affect the explainability, and how does the method address this limitation?
>
> **A:** We would like to clarify that we do not directly treat subtrees as the global subgraph concepts. Instead, we take the subgraphs extracted with the help of subtrees as the explanations. As illustrated in Fig. 2, we identify overlapping subgraph patterns among subtrees with similar embeddings, where similar embeddings typically indicate shared critical concepts for classification. As a result, we do not overlook finer subgraph features, such as specific motifs or relational patterns [1][2].
>
> > **Q:** Given the similarities in last-layer embeddings, k-means may face issues that embeddings can be smooth or overlapping. As a result, it might not distinguish nuanced structural variations effectively. Did you evaluate whether k-means clustering accurately captures meaningful and distinct concepts in the last-layer embeddings? Do you consider alternative clustering approaches for finer clustering?
>
> **A:** We would like to clarify that we have leveraged the property that "last-layer embeddings are smooth or overlapping" in designing our algorithm, rather than treating it as a drawback. In a graph, when several nodes have similar embeddings, it is likely that they collectively contribute to the same concept or prominent pattern. As shown in Fig. 2, we use a clustering algorithm to identify which subtrees contribute to the same concept and find the overlapping subgraph patterns among these subtrees, thereby extracting subgraph concepts. This approach accelerates the subgraph mining process.
>
> Figs. 3 and 5 demonstrate that k-means has effectively identified meaningful and distinct concepts in our experiments. Moreover, the superior fidelity performance presented in Tables 1 and 2 confirms that our method accurately captures subgraph concepts. It is important to note that the clustering algorithm is neither the focus nor the main contribution of our paper. Instead, our work addresses the complexity challenge of identifying critical subgraph concepts at the global level.
>
> Our contributions include several novel designs, such as utilizing subtrees to effectively and efficiently extract subgraph patterns and employing the root node embedding as the subtree embedding to further reduce computational costs. Given that TreeX can extract global subgraph patterns while existing global-level methods cannot, it achieves comparable and even superior performance to existing local-level approaches, offering both global- and local-level explainability.
>
> > **Q:** Discussion and evaluation on more architectures (GCN, GraphSAGE, GAN, etc), as different GNN architectures have varying expressive power and aggregation mechanisms.
>
> **A:** We primarily evaluate our method on GIN for two reasons. First, GIN is the most powerful 1-WL-based GNN and has efficiency comparable to other GNN variants, making it a more common choice for solving problems with GNNs. Second, almost all existing GNN explainability methods evaluate only a single variant of GNNs. However, we provide detailed discussions and proofs in Appendix A on how our method can explain less powerful GNNs.
> In summary, since less powerful GNNs have lower discriminative power than GIN, we employ a hash model to aid in distinguishing global graph concepts. This model is designed to handle subtrees with similar node distributions or node sets but significantly different structures. As a result, our TreeX approach remains applicable when explaining those GNN variants.

---

### Official Review · Reviewer_JEW3 · 2024-11-04

**Soundness:** 2
**Presentation:** 2
**Contribution:** 1
**Rating:** 3
**Confidence:** 5

**Summary:**

This paper proposes a type of global GNN explanation methods, by extracting the subtrees incurred by the GNN message passing mechanisms. The paper argues that this can provide more intuitive local, class, and global level explanations.

**Strengths:**

- The proposed method is intuitive and technically sound.
- The experimental analysis is abundant.

**Weaknesses:**

- First, the definitions for different types of GNN explanations are quite vague. This paper claims that GNN explanation methods can be categorized into local and global levels; it claims that instance-level explanation is local-level. Under this definition, the proposed method should be considered as local-level, since it only works in the instance-level graph graph classification task where all the subtrees are extracted from the graph to be predicted.
- Second, the proposed methods only seem to work with graph-level prediction tasks and do not seem to work with node-level prediction tasks. I did not find the paper explicitly discussing that. The existing popular GNN explainability methods, such as GNN Explainer, can work with both node and graph-level prediction tasks. The paper should clearly emphasize its limitations.
- A key idea of this paper - last layer node embedding represents the full L-hop subtrees, is a well-known result in the GNN research domain (for example, it has been taught throughout the Stanford CS224W course with millions of views on Youtube). The justifications in Section 4.2 look redundant to me. Furthermore, Theorem 4.2 and its proof are a direct application of the results from the GIN paper, which is also redundant.
- Overall, I do not find the proposed method offers a new understanding of the explainable GNN domains, especially given that it can only work with graph-level prediction tasks, and the performance is only on par or even worse than existing methods (Table 1). I am happy to defend my opinion further if needed

**Questions:**

- Has the authors considered a more basic baseline, where we can use an attention-based aggregator over last layer node embeddings to make the final graph level prediction? The, we can simply return the top-k subtrees based on the attention scores. I found this method to be more convenient, easier to implement, and more efficient to compute. I did not get how the proposed approach fundamentally differs from that simpler baseline.

---

> ### Author Response · Authors · 2024-12-02
>
> Thank you for recognizing the clarity and technical soundness of our paper, and the thoroughness and comprehensiveness of the experiments. For the weaknesses you raised:
>
> > **Q:** I think the approach proposed is a local-level explainer because it only works on ``instance-level graph classification tasks''.
>
> **A:** We did not propose a grouping criterion for graph classification tasks. Therefore, we didn't define the term of ``instance-level graph classification task'' . Instead, we focus on explaining the graph classification task, and provide both local (instance)-level and global(dataset)-level explanations by identifying crucial subgraph patterns.
>
> For instance-level explanations, we would be able to directly identify which subgraphs contribute more to the target class, like other local-level explainers. For dataset-level explanations, we would obtain a rule similar to the one in Fig 1 (in real case, the rule can be very long, with all the outstanding patterns for the dataset presented in the rule). For example, a rule with 30 subgraph patterns can be: 0.4\*Pattern 1 + 0.2\*Pattern 2 - 1.0\*Pattern 3 + 0.5\*Pattern 4 + ... + 0.8\*Pattern 30. This rule tells us that, for instance, Pattern 30 is important to the task (coefficient=0.7), and Pattern 3 is not important to the task (coefficient=-1.0). Graphs with more positive patterns would be more likely classified to this target class. This gives a comprehensive view of how GNNs make predictions for this task, over the entire dataset.
>
> Uncovering crucial subgraph patterns in a dataset is challenging due to the vast search space for subgraphs, a problem that no previous methods have successfully addressed. In our work, we tackle this by collecting subtrees, which significantly reduce the search space, and then constructing the subgraph patterns afterward. While this may seem like a simple solution, no prior work has proposed this approach, making it a novel and impactful contribution to solving this challenge—much like Gauss's summation formula, which appeared straightforward only after he introduced it.
>
> > **Q:** The proposed methods only works with graph-level prediction not node-level prediction tasks, unlike GNNExplainer that produces instance-level explanations for both node and graph classification.
>
> **A:** Yes, we focus on a different problem than GNNExplainer. Their goal is to generate node importance for individual graph instances to explain predictions for those specific instances. However, we address a different challenge: in graph classification tasks with numerous graphs in a dataset, uncovering dataset-level crucial subgraph patterns becomes difficult due to the vast search space. This is why our focus is on graph classification. Thank you for your suggestion. We have clarified in our revised manuscript that we do not focus on node classification.
>
> > **Q:** A key idea of this paper - last layer node embedding represents the full L-hop subtrees, is a well-known result in the GNN research domain. Redundant statements and theorem.
>
> **A:** The idea may seem straightforward, but no one before us has proposed leveraging it for extracting global-level subgraph explanations. This can be similar to the Chain-of-Thought (CoT) prompting in NLP—while the concept might appear simple in hindsight, its significant impact was only recognized after it was introduced. Likewise, our approach represents a novel application of this concept in the context of subgraph explanations. Similarly, you might feel that the statements and theorem are too obvious or simply a direct application of network knowledge or previous work. However, this paper is written for a broad audience with diverse backgrounds, including readers who may have less expertise in this domain. For them, these statements, theorems, and their proofs are essential to ensure clarity and understanding.

---

> > ### Author Response · Authors · 2024-12-02
> >
> > > **Q:** Comments on novelty. The performance is only on par or even worse than existing methods (Table 1).
> >
> > **A:** We want to emphasize that our focus is not on identifying critical nodes, edges, or subgraphs for GNN predictions on individual graph instances, as is the case with most existing GNN explainability methods. Instead, our goal is to address the complexity challenge of extracting dataset- or task-level critical subgraph patterns across a large number of graph instances. This complexity challenge typically arises in graph classification tasks. To tackle this, we have introduced several novel designs that have not been applied in the related literature, including the utilization of subtrees to extract subgraph patterns and using root node embeddings as subtree embeddings. Our method produces both global- and local-level explanations for GNNs, whereas most existing works focus on only one aspect.
> >
> > The performance is NOT on par with or worse than existing methods, as these methods do not generate clear global-level subgraph explanations at all. In Table 1, we demonstrate that TreeX can effectively provide instance-level explanations as well, achieving comparable or even superior performance to existing instance-level approaches.
> >
> > > **Q:** Has the authors considered a more basic baseline, where we can use an attention-based aggregator over last layer node embeddings to make the final graph level prediction? The, we can simply return the top-k subtrees based on the attention scores.
> >
> > **A:** The problem we focus on is extracting dataset- or task-level critical subgraph patterns across a large number of graph instances, rather than identifying critical nodes within each graph instance as the reviewer suggested. This is why we did not include a baseline like the one you described. For comparing instance-level explanations, I have chosen more advanced methods such as PGExplainer, SubgraphX, and Eig-Search as baselines, which are significantly more powerful than attention-based approaches. Our method achieves comparable or even superior performance to these existing instance-level approaches.

---

### Official Review · Reviewer_oevP · 2024-11-04

**Soundness:** 2
**Presentation:** 2
**Contribution:** 2
**Rating:** 5
**Confidence:** 4

**Summary:**

This paper proposes a method for providing global explanations in GNNs, specifically targeting maximally powerful MPGNNs. The approach leverages clustering based on node embeddings within MPGNNs to explain substructure information at a global level. Its effectiveness is evaluated across various datasets.

**Strengths:**

- Providing global explanations is a crucial aspect in the development of trustworthy GNN models.
 - The paper provides adequate background information through preliminaries and related sections.
 - The effectiveness of the proposed method is evaluated by multiple datasets.

**Weaknesses:**

- The methodological details in the paper are unclear. For example, while it mentions identifying graph substructures based on node embeddings, the specific approach is not adequately detailed, making it challenging to understand.
 - A significant issue is the limited evaluation measures, with insufficient justification provided. Various measures, such as sparsity, fidelity (+ and -), and fidelity\delta [1], are commonly used and could be considered in the evaluation.
 - More comparisons with existing global XAI methodologies are needed. Methods like D4Explainer [2] and TAGE [3], which also can provide global explanations, would enhance the baseline comparisons.

References:

[1] Zheng et al., "Towards Robust Fidelity for Evaluating Explainability of Graph Neural Networks," ICLR 2024.

[2] Chen et al., "D4Explainer: In-distribution Explanations of Graph Neural Network via Discrete Denoising Diffusion," NeurIPS 2023.

[3] Xie et al., "Task-Agnostic Graph Explanations," NeurIPS 2022.

**Questions:**

Please refer to the weaknesses above.

---

> ### Author Response · Authors · 2024-12-02
>
> > **Q:** The methodological details in the paper are unclear. For example, while it mentions identifying graph substructures based on node embeddings, the specific approach is not adequately detailed, making it challenging to understand.
>
> **A:** The identification of graph substructures based on node embeddings is a core contribution of our method, and we provide a detailed approach to clarify this process. Specifically, we use the root node embedding as the representation of its corresponding \(L\)-hop subtree embedding. Within each graph instance, we cluster these subtree embeddings based on the principle that similar embeddings represent similar concepts, aiming to extract meaningful subgraph concepts. After clustering, we identify the overlapping subgraph structures among the subtrees in each cluster and designate these as the critical subgraph concepts underlying the cluster. This approach ensures that the identified concepts are both interpretable and representative of the underlying graph structures. By combining node embedding-based representation with clustering and overlap detection, our method effectively uncovers key substructures within graph instances, addressing the need for detailed methodological clarity.
>
> > **Q:** A significant issue is the limited evaluation measures, with insufficient justification provided. Various measures, such as sparsity, fidelity (+ and -), and fidelity $\delta$ [1], are commonly used and could be considered in the evaluation.
>
> **A:** In our study, we focused on metrics directly aligned with the objectives of our method, namely AccFidelity and ProbFidelity, which correspond to **Fid-** and evaluate the alignment of explanations with the model’s predictions. These metrics are particularly relevant for assessing the quality of explanations in GNNs and are split into accuracy-based and probability-based evaluations for greater granularity.
>
> Metrics such as **Fidelity** $\delta$ and **Fid+** are indeed designed for explainers that evaluate the impact of removing important features. However, our method goes beyond simply identifying and removing features—it associates weights with subgraph concepts and constructs linear combinations of these concepts as explanations. This unique design makes our method less suited to evaluations based on feature removal, as these weights are integral to the explanations and their predictive utility.
>
> To address concerns about justification, we will include this clarification in our manuscript to better explain the rationale behind our choice of metrics. By evaluating our method using AccFidelity and ProbFidelity, we effectively validate the fidelity and interpretability of our explanations. Our results demonstrate high performance on these metrics, further highlighting the robustness and effectiveness of our approach in providing meaningful and class-specific explanations.
>
> > **Q:** Methods like D4Explainer [2] and TAGE [3], which also can provide global explanations, would enhance the baseline comparisons.
>
> **A:** Our method focuses on producing **factual**, **class-specific** global explanations for entire datasets, similar to the baselines compared in our paper. We achieve this by extracting global subgraph concepts and constructing linear combinations of these concepts for each class. In contrast, TAGE provides task-agnostic explanations for GNN embeddings, targeting self-supervised **multitask** models without specific downstream tasks. D4Explainer generates in-distribution explanations by learning graph distributions, with an emphasis on **counterfactual** and model-level insights. While both methods are valuable, their objectives differ significantly from ours, making direct baseline comparisons less relevant. To address this feedback, we will add a discussion of both TAGE and D4Explainer to the related work section of our manuscript, highlighting their contributions and clarifying how our approach differs in scope and objectives.

---

### Meta-Review · Area_Chair_fZfe · 2024-12-19

**Metareview:**

Based on the reviewers' comments and my own reading, this paper presents TreeX, a method for explaining Graph Neural Networks through critical subtree extraction. The approach leverages message passing mechanisms to extract subtree information and utilizes last-layer node embeddings to represent subtree features. The authors claim that TreeX provides intuitive graphical explanations at local, class, and global levels for Message-Passing GNNs, validating its effectiveness across multiple datasets.

The paper demonstrates several notable strengths. It introduces an innovative perspective by using subtrees rather than nodes or linguistic rules to explain global GNN behavior. The method achieves computational efficiency by leveraging message passing mechanisms and last-layer embeddings for subtree feature extraction, thereby avoiding time-consuming subgraph enumeration processes. Furthermore, the approach provides comprehensive interpretability through explanations at multiple levels, and the authors have conducted extensive experimental validation across various datasets.

However, the paper suffers from several significant weaknesses that ultimately undermine its contribution. The methodological details are insufficiently explained, with multiple reviewers noting that crucial steps, such as the specific approach for identifying graph substructures based on node embeddings, lack adequate explanation. The evaluation framework is incomplete, missing important metrics such as sparsity and various fidelity measures that are standard in the field. Additionally, the method appears to be limited to graph-level prediction tasks, a significant restriction that is not explicitly acknowledged or discussed in the paper.

The theoretical innovation of the work is also questionable. Several core ideas, such as the representation of L-hop subtrees by last-layer node embeddings, are well-established results in the GNN domain. The paper lacks comprehensive comparisons with existing methods like D4Explainer and TAGE. Technical concerns have also been raised about the global concept extraction method, particularly regarding embedding space alignment across different graph instances and the discriminative power of the frequency-based class rule generation approach.

Given these limitations, I recommend rejection of this paper. While the work presents interesting ideas and shows potential practical value, it requires substantial improvements in several areas before it can make a meaningful contribution to the field. The authors should provide more detailed explanations of key methodological steps, extend their evaluation framework, include comparisons with recent related work, address the limitations regarding node-level predictions, and provide stronger theoretical justification for their approach. With these improvements, the paper could potentially make a valuable contribution to the field of explainable GNNs.

**Additional Comments On Reviewer Discussion:**

The rebuttal period focused on three main concerns raised by the reviewers, with the authors providing detailed responses to each point.

First, regarding the unclear methodological details, particularly about identifying graph substructures from node embeddings, the authors provided a more detailed explanation of their approach. They clarified that they use root node embeddings to represent (L)-hop subtree embeddings, followed by clustering similar embeddings to extract meaningful concepts. The authors explained that they identify overlapping subgraph structures among subtrees in each cluster to determine critical subgraph concepts. This response helps address the clarity concerns, though the technical details could have been better presented in the original manuscript.

Second, concerning the limited evaluation measures, the authors defended their choice of metrics (AccFidelity and ProbFidelity) by explaining that these metrics directly align with their method's objectives. They argued that traditional metrics like Fidelity∆ and Fid+ are less suitable for their approach because their method uses weighted subgraph concepts rather than simple feature removal. While this explanation provides some justification for their evaluation choices, the lack of comprehensive evaluation metrics still remains a limitation of the work.

Third, regarding the comparison with methods like D4Explainer and TAGE, the authors argued that these methods have fundamentally different objectives from their work. They explained that while their method focuses on factual, class-specific global explanations, TAGE provides task-agnostic explanations for GNN embeddings, and D4Explainer generates in-distribution explanations with different goals. The authors proposed adding a discussion of these methods to their related work section. This response partially addresses the concern about baseline comparisons, though a broader experimental comparison would have strengthened the paper.

In weighing these points, while the authors' responses provide helpful clarifications, they do not fully address the core limitations identified in the initial reviews. The methodology explanation helps but should have been in the original manuscript, the justification for limited evaluation metrics remains partially convincing, and the lack of comprehensive baseline comparisons still weakens the paper's contribution. Combined with other concerns about theoretical innovation and limited applicability raised in the initial reviews, these responses do not sufficiently strengthen the paper to overcome the recommendation for rejection.

---

### Decision · Program_Chairs · 2025-01-22

Reject